# ON DISCRETE SYMMETRIES OF ROBOTICS SYSTEMS: A GROUP-THEORETIC AND DATA-DRIVEN ANALYSIS

## ABSTRACT

In this work, we study the Morphological Symmetries of dynamical systems with one or more planes of symmetry, a predominant feature in animal biology and robotic systems, characterized by the duplication and balanced distribution of body parts. These morphological symmetries imply that the system's dynamics are symmetric (or approximately symmetric), which in turn imprints symmetries in optimal control policies and in all proprioceptive and exteroceptive measurements related to the evolution of the system's dynamics. For data-driven methods, symmetry represents an inductive bias that justifies data augmentation and the construction of symmetric function approximators. To this end, we use Group Theory to present a theoretical and practical framework allowing for (1) the identification of the system's morphological symmetry Group $\mathcal{G}$, (2) the characterization of how the group acts upon the system state variables and any proprioceptive and exteroceptive measurement, and (3) the exploitation of data symmetries through the use of $\mathcal{G}$-equivariant/$\mathcal{G}$-invariant Neural Networks, for which we present experimental results on synthetic and real-world applications, demonstrating how symmetry constraints lead to better sample efficiency and generalization while reducing the number of trainable parameters.

## 1 INTRODUCTION

Symmetries are a predominant feature in animal biology. The majority of living (and extinct) species are bilaterally or radially symmetric (i.e., having one or more planes of symmetry), a property intuitively recognized by the patterns of balanced distribution and duplication of body parts and shapes (Holló, 2017). Likewise, most robotic systems are symmetric, often featuring more precise symmetries than nature due to the accurate duplication of body parts and the tendency to design mechanisms with symmetric volumes and mass distributions. These morphological symmetries of animals and robots imply that the dynamics and control of body motions are also approximately symmetric, resulting in all proprioceptive and exteroceptive measurements, related to the evolution of the system's dynamics (e.g. joint torques, depth images, contact forces), to be also symmetric. This highly relevant inductive bias is frequently left unexploited in most data-driven applications in the fields of robotics, computer graphics, computational biology, and control.

Recent works in computer graphics (Yeh et al., 2019; Abdolhosseini et al., 2019; Yu et al., 2018) and robotics/dynamical systems (Van der Pol et al., 2020; Ordonez-Apraez et al., 2022; Hamed & Grizzle, 2013; Finzi et al., 2021a) have exploited through different approaches the morphological symmetry group associated with bilateral (or sagittal) symmetry (the reflection group $\mathcal{C}_2$), obtaining improvements in generalization and sample efficiency of function approximators. Notably, Zinkevich & Balch (2001) proved that Markov Decision Processes with state symmetries have symmetric optimal value and policy functions. Despite these encouraging contributions, exploiting the inductive bias of morphological symmetries is not a widespread technique in the research community. We attribute the scarce adoption of these techniques to the lack of a unifying theoretical and practical framework, allowing to identify different morphological symmetries in arbitrary dynamical systems and efficiently and conveniently exploit them in data-driven applications.

This work takes a step towards this unifying framework by studying morphological symmetries through the lens of dynamical systems and group theory[1]. Our theoretical contributions are:

---

[1]The field of mathematics that studies symmetries, which is broadly used in Machine Learning (ML)

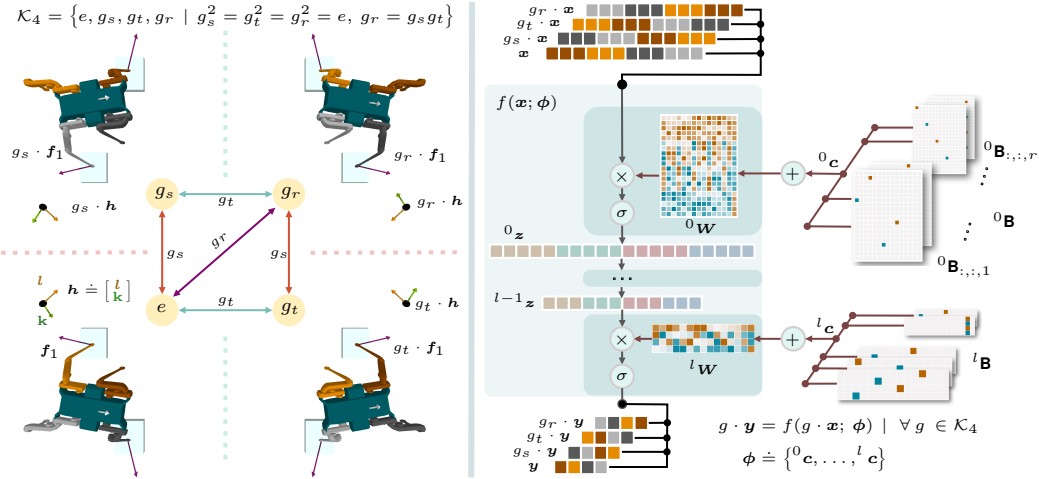

Figure 1: **Left:** Caley diagram and top-view (see 3D animation) of symmetric configurations of the quadruped robot Solo, whose morphological symmetries (described by the Klein four-group $\mathcal{K}_4$) allow it to imitate the effect of reflections ($g_s, g_t$) and $180°$ rotations of space ($g_r$). Transformations affect both proprioceptive (state space, CoM linear $l$ and angular $\mathbf{k}$ momentum) and exteroceptive (terrain elevation, external disturbances) quantities. **Right:** Diagram of a $\mathcal{K}_4$-equivariant NN. Each of the layer's linear maps $\boldsymbol{W}$ is constructed as a weighted average of the basis of the space of equivariant linear maps $\mathbf{B}$, computed from the $\mathcal{K}_4$ symmetries of the input-output spaces (see section 5).

❊ A group-theoretic formalization of the concept of discrete morphological symmetry.
❊ A characterization of how the morphological symmetry group $\mathcal{G}$ affects the system's state variables and any relevant proprioceptive and exteroceptive measurements. Facilitating the identification of $\mathcal{G}$ and the augmentation of proprioceptive and exteroceptive measurements.

Once the morphological symmetry group $\mathcal{G}$ is identified, our practical contributions focus on the efficient construction and versatile use of $\mathcal{G}$-equivariant neural networks, for arbitrary discrete morphological symmetry groups $\mathcal{G}$, for which we:

❖ Derived an optimal initialization for the trainable parameters of equivariant layers[2].
❖ Demonstrate that $\mathcal{G}$-equivariance reduces the trainable parameters by approximately $1/|\mathcal{G}|$.
❖ Enable the construction of large scale $\mathcal{G}$-equivariant networks by mitigating the construction computational complexity and the storage memory complexity of equivariant architectures[2].

## 2 BACKGROUND ON SYMMETRY GROUPS

In a nutshell, a symmetry group in Group Theory is an abstraction of the concept of symmetries that different geometric objects might have, understanding symmetry as a transformation that when applied to an object conserves a relevant property of its structure. For instance, in fig. 1-left the Klein four-group $\mathcal{K}_4$ describes the symmetries that vectors, pseudo-vectors, rigid bodies, and a quadruped robot have to $180°$ rotations ($g_r$) and two perpendicular reflections ($g_s, g_t$). Transformations that preserve vector magnitudes and energy. While on fig. 1-right the same group describes the symmetries of vector spaces, representing the quadruped robot's state $\boldsymbol{x}$ and legs contact state $\boldsymbol{y}$.

Formally, a symmetry group is a set of invertible symmetry transformations (or actions) $\mathcal{G} = \{e, g_1, g_2, \dots\}$, containing the trivial action $e$ (which leaves objects unchanged) and having a binary operator $(\cdot) : \mathcal{G} \times \mathcal{G} \to \mathcal{G}$, that is associative (i.e. $g_1 \cdot (g_2 \cdot g_3) = (g_1 \cdot g_2) \cdot g_3$), which composes group members into other group members, such as $g_r = g_s \cdot g_t$ for $\mathcal{K}_4$ (see fig. 1). Group representations are characterizations of how each action $g$ transforms a specific geometric object, say $\boldsymbol{x} \in \mathbb{R}^k$. A representation $\rho_{\boldsymbol{x}} : \mathcal{G} \to GL(k)$ ($GL$ : General Linear group) is a group homomorphism associating each $g$ to an invertible linear map $\rho_{\boldsymbol{x}}(g) \in \mathbb{R}^{k \times k}$ specifying how the object $\boldsymbol{x}$ is transformed, that is: $g(\boldsymbol{x}) \equiv g \cdot \boldsymbol{x} \doteq \rho_{\boldsymbol{x}}(g)\boldsymbol{x}$. Since group actions are abstract, it is common to define different object-dependent representations for each action, as we will see throughout this work.

---

[2] The link to an anonymous repository is available in the official comments on OpenReview, accessible to reviewers and area chairs. The repository will be made public to the general public upon paper acceptance.

A fundamental concept for this work is the notion of function $\mathcal{G}$-equivariance and $\mathcal{G}$-invariance. Consider the function $f : \mathbb{R}^n \to \mathbb{R}^m$. $f$ is said to be $\mathcal{G}$-equivariant or $\mathcal{G}$-invariant if:

$$\underbrace{g \cdot \boldsymbol{y} = f(g \cdot x) \quad | \quad \forall \quad g \in \mathcal{G}}_{Equivariance} \qquad \underbrace{\boldsymbol{y} = f(g \cdot x) \quad | \quad \forall \quad g \in \mathcal{G},}_{Invariance} \tag{1}$$

In words, an equivariant function maps symmetries of the input to symmetries of the output, while an invariant function maps symmetries of the input to an invariant output.

Being this short section undoubtedly an unsatisfactory introduction to Group Theory, we refer the uninitiated and interested reader to Bronstein et al. (2021) for a remarkable introduction to the field.

## 3 LAGRANGIAN MECHANICS AND SYMMETRIES OF DYNAMICAL SYSTEMS

First, we provide a group-theoretic perspective of symmetries in a dynamical system. To this end, let us consider a dynamical system with generalized coordinates $\boldsymbol{q} \in \mathrm{Q} \subseteq \mathbb{R}^n$ and velocities $\dot{\boldsymbol{q}} \in \mathrm{T}_{\boldsymbol{q}}\mathrm{Q} \subseteq \mathbb{R}^n$; as well as a Lagrangian function $\mathcal{L} : \mathrm{Q} \times \mathrm{T}_{\boldsymbol{q}}\mathrm{Q} \to \mathbb{R} = \mathcal{T}(\boldsymbol{q}, \dot{\boldsymbol{q}}) - \mathcal{U}(\boldsymbol{q}, \dot{\boldsymbol{q}})$. Being Q the constrained configuration space, $\mathrm{T}_{\boldsymbol{q}}\mathrm{Q}$ the configuration tangent space at $\boldsymbol{q}$ (i.e., the space of generalized velocities), and $\mathcal{T}(\boldsymbol{q}, \dot{\boldsymbol{q}}), \mathcal{U}(\boldsymbol{q}, \dot{\boldsymbol{q}})$ the state kinetic and potential energies, respectively.

The symmetries of a dynamical system are defined as transformations in the space of generalized coordinates that keep the energy state of the system unchanged (Ostrowski & Burdick, 1996). In this work, we study time-invariant point-transformations[3] of generalized coordinates $g : \mathrm{Q} \to \mathrm{Q}$, which are interpreted as actions of a symmetry group, i.e. $g \in \mathcal{G}$. Denoting $\rho_{\mathrm{Q}}(g) \in \mathbb{R}^{n \times n}$ as the action representation in Q, we define the transformed coordinates as $g(\boldsymbol{q}) \doteq \rho_{\mathrm{Q}}(g)\boldsymbol{q} = g \cdot \boldsymbol{q}$. Consequently, the velocity and acceleration of the transformed coordinates are given by $g \cdot \dot{\boldsymbol{q}}$ and $g \cdot \ddot{\boldsymbol{q}}$, respectively, considering that $\frac{dg(\boldsymbol{q})}{dt^k} = \frac{\partial g(\boldsymbol{q})}{\partial \boldsymbol{q}} \frac{d\boldsymbol{q}}{dt^k} = \rho_{\mathrm{Q}}(g)\frac{d\boldsymbol{q}}{dt^k} \doteq g \cdot \frac{d\boldsymbol{q}}{dt^k}$.

Formally, we say that a dynamical system has a symmetry group $\mathcal{G}$ if its Lagrangian is $\mathcal{G}$-invariant:

$$\mathcal{L}(\boldsymbol{q}, \dot{\boldsymbol{q}}) = \mathcal{L}(g \cdot \boldsymbol{q}, g \cdot \dot{\boldsymbol{q}}) \quad | \quad \forall \quad g \in \mathcal{G}, \quad \boldsymbol{q} \in \mathrm{Q}, \quad \dot{\boldsymbol{q}} \in \mathrm{T}_{\boldsymbol{q}}\mathrm{Q}. \tag{2}$$

Because the Lagrangian structure differs between the original $(\boldsymbol{q}, \dot{\boldsymbol{q}})$ and transformed coordinates $(g \cdot \boldsymbol{q}, g \cdot \dot{\boldsymbol{q}}) \mid \forall \, g \in \mathcal{G}$, when we derive the Equations of Motion (EoM) of the system in the transformed coordinates, we obtain a set of EoMs describing the true system dynamics in different coordinate systems. Formally, if we derive the EoM through the Euler-Lagrange equation of the second order $\left( \frac{d}{dt} \frac{\partial \mathcal{L}(\boldsymbol{q}, \dot{\boldsymbol{q}})}{\partial \dot{\boldsymbol{q}}} - \frac{\partial \mathcal{L}(\boldsymbol{q}, \dot{\boldsymbol{q}})}{\partial \boldsymbol{q}} \equiv \mathbf{M}(\boldsymbol{q})\ddot{\boldsymbol{q}} - \boldsymbol{\tau}(\boldsymbol{q}, \dot{\boldsymbol{q}}) = \mathbf{0} \right)$, the distinct EoM are equivariant[4] to each other (Lanczos, 2020), a property we will refer to as dynamics $\mathcal{G}$-equivariance:

$$g \cdot [\underbrace{\mathbf{M}(\boldsymbol{q})\ddot{\boldsymbol{q}}}_{Inertial} - \underbrace{\boldsymbol{\tau}(\boldsymbol{q}, \dot{\boldsymbol{q}})}_{Moving}] = \underbrace{\mathbf{M}(g \cdot \boldsymbol{q})g \cdot \ddot{\boldsymbol{q}}}_{Inertial} - \underbrace{\boldsymbol{\tau}(g \cdot \boldsymbol{q}, g \cdot \dot{\boldsymbol{q}})}_{Moving} = \mathbf{0} \mid \forall g \in \mathcal{G}, \, \boldsymbol{q} \in \mathrm{Q}, \, \dot{\boldsymbol{q}} \in \mathrm{T}_{\boldsymbol{q}}\mathrm{Q}. \tag{3}$$

Denoting $\mathbf{M}(\boldsymbol{q}) : \mathrm{Q} \to \mathbb{R}^{n \times n}$ as the generalized mass matrix function and $\boldsymbol{\tau}(\boldsymbol{q}, \dot{\boldsymbol{q}}) : \mathrm{Q} \times \mathrm{T}_{\boldsymbol{q}}\mathrm{Q} \to \mathbb{R}^n$ as the generalized moving forces at $\boldsymbol{q}$ and $\dot{\boldsymbol{q}}$. Note that, in eq. (3) the original and transformed dynamics are related linearly by the Jacobian of the coordinate transformation $\partial g(\boldsymbol{q})/\partial \boldsymbol{q} = \rho_{\mathrm{Q}}(g)$ (Wheeler, 2014), which to preserve notation is reduced to $g$.

To ensure dynamics $\mathcal{G}$-equivariance (eq. (3)), both the generalized inertial and moving forces need to be independently equivariant, meaning:

$$\mathbf{M}(g \cdot \boldsymbol{q}) = g\mathbf{M}(\boldsymbol{q})g^{-1} \quad \wedge \quad g \cdot \boldsymbol{\tau}(\boldsymbol{q}, \dot{\boldsymbol{q}}) = \boldsymbol{\tau}(g \cdot \boldsymbol{q}, g \cdot \dot{\boldsymbol{q}}) \quad | \forall g \in \mathcal{G}, \, \boldsymbol{q} \in \mathrm{Q}, \, \dot{\boldsymbol{q}} \in \mathrm{T}_{\boldsymbol{q}}\mathrm{Q}. \tag{4}$$

The equivariance of the generalized mass matrix provides a pathway for the identification of the symmetry group $\mathcal{G}$, and the group action representations $\rho_{\mathrm{Q}}(g) \mid g \in \mathcal{G}$ (see section 4.2). While the equivariance of the generalized moving forces (which in practice usually incorporates control forces, constraint forces, and external interactions) implies that dynamics $\mathcal{G}$-equivariance is held until a symmetry braking force violates the equivariance of $\boldsymbol{\tau}$.

**Floating-base robotic/dynamical systems**: Let us now narrow our focus to floating-based dynamical systems. Namely, legged/flying/swimming robots, animals, and animated characters evolving in

---

[3]A point-transformation $g$ is a finite, invertible, continuous, and differentiable function of $\boldsymbol{q}$(Lanczos, 2020)
[4]Some authors refer to this property as *covariance* of the EoMs (Wheeler, 2014; Lanczos, 2020)

a Euclidean space of $d$ dimensions (with its corresponding Euclidean Lie Group $\mathbb{E}_d$), whose generalized coordinates can be decoupled into $\boldsymbol{q} = \begin{bmatrix} \boldsymbol{X}_B \\ \hat{\boldsymbol{q}} \end{bmatrix} \in Q \doteq \mathbb{E}_d \times Q_J$.[5] Where $\boldsymbol{X}_B \in \mathbb{E}_d$ represents the system's base (or center of mass (CoM)) position and orientation[6]. $\hat{\boldsymbol{q}} \in Q_J \subseteq \mathbb{R}^{n_J}$ represents the internal Degrees of Freedom (DoF) constrained configuration. And $Q_J$ the internal configuration space commonly referred to as joint space. In this coordinate space, we can differentiate the effect of $g$ on $\mathbb{E}_d$ and $Q_J$, noting that: $g(\boldsymbol{q}) = g \cdot \boldsymbol{q} = \rho_Q(g)\boldsymbol{q} = \begin{bmatrix} \rho_{\mathbb{E}_d}(g) & \boldsymbol{0} \\ \boldsymbol{0} & \rho_{Q_J}(g) \end{bmatrix} \begin{bmatrix} \boldsymbol{X}_B \\ \hat{\boldsymbol{q}} \end{bmatrix} \mid \forall\, g \in \mathcal{G}$. Representing $\rho_{\mathbb{E}_d}(g) \in \mathbb{E}_d$ a homogeneous matrix transformation affecting the base, and $\rho_{Q_J}(g) \in \mathbb{R}^{n_J \times n_J}$ a transformation on the joint-space. The differentiation becomes handy in identifying the system's continuous and discrete (section 4) symmetries.

**Continuous symmetries of floating-base systems**: The most commonly studied and exploited symmetries of floating-based systems are the continuous symmetries of the Euclidean space in which the system evolves, i.e., symmetry actions $\overline{g} \in \mathbb{E}_d$, involving $d$-dimensional rotations/reflections + translations[7]. The property of these actions $\overline{g}$, which is of most interest to us, is the $\mathbb{E}_d$-invariance of the joint-space configuration: $\overline{g} \cdot \hat{\boldsymbol{q}} = \hat{\boldsymbol{q}} \iff \rho_{Q_J}(\overline{g}) = \boldsymbol{I}_{n_J} \mid \forall\, \overline{g} \in \mathbb{E}_d$.

## 4 DISCRETE MORPHOLOGICAL SYMMETRIES (DMSS)

A Discrete Morphological symmetry (DMS) is a mathematical formalization of the intuitive property of floating-base dynamical systems that can imitate the effect of rotations, translations, and infeasible reflections of space with feasible discrete change in the system configuration. To better introduce the concept of DMS it is useful to first study the most simple (and most frequent) instance of a DMS: the reflection symmetry, which all humans and most animals approximately possess (Holló, 2017).

**Reflection DMS $\mathcal{G} = \mathcal{C}_2$**: Despite most floating-base dynamical systems being symmetric w.r.t reflections of space ($\overline{g} \in \mathbb{E}_d$), in practice, it is common to ignore these reflection symmetries, since, in general, it is impossible to subject a real-world robotic/dynamical system to a true reflection of space (Selig, 2005). Think of your own body as a floating-base system, you can move and rotate your base (hip) in space but you are unable to execute a true reflection of space, which will force your heart to switch sides (and certainly die). Fortunately, your body is symmetric w.r.t. the sagittal plane (we will assume perfect symmetry for now) which allows you to imitate the effect of a true reflection of space by modifying your internal configuration (your body pose), and rotating and translating your base (i.e., with a feasible discrete change in your configuration, see supp.fig 6a). Therefore, you have a discrete morphological symmetry associated with the reflection group $\mathcal{C}_2$.

**Discrete Morphological Symmetries with higher order groups:** When the finite symmetry group $\mathcal{G}$ has group order $|\mathcal{G}| > 2$, a DMS can imitate both rotations/reflections and translations in $\mathbb{E}_d$. Therefore, the group $\mathcal{G}$ will be isomorphic to one of the groups in Euclidean geometry. Most frequently $\mathcal{G}$ is a Cyclic $\mathcal{C}$ or Dihedral $\mathcal{D}$ group. See examples for $\mathcal{C}_2$ on supp.figs 5 and 6a, for $\mathcal{C}_3$ in supp.fig 5, and for $\mathcal{D}_4 \equiv \mathcal{K}_4$ in fig. 1.

**Definition of Discrete Morphological Symmetry**: Consider a floating-base dynamical system, with generalized coordinates $\boldsymbol{q} \in Q \doteq \mathbb{E}_d \times Q_J$, evolving in a $d$-dimensional Euclidean space. The system is said to have a DMS if, for a given continuous symmetry action $\overline{g} \in \mathbb{E}_d$, there exists an action $g \in \mathcal{G}$, that is proper ($|\rho_{\mathbb{E}_d}(g)| = 1$) and non-trivial in joint-space ($\rho_{Q_J}(g) \neq \boldsymbol{I}_d$), such that:

$$\mathcal{L}(\boldsymbol{q}, \dot{\boldsymbol{q}}) = \mathcal{L}(\overline{g} \cdot \boldsymbol{q}, \overline{g} \cdot \dot{\boldsymbol{q}}) = \mathcal{L}(g \cdot \boldsymbol{q}, g \cdot \dot{\boldsymbol{q}}) \quad \forall\, \boldsymbol{q} \in Q,\ \dot{\boldsymbol{q}} \in T_{\boldsymbol{q}}Q,\ g \in \mathcal{G},\ \overline{g} \in \mathbb{E}_d. \quad (5)$$

Where $\overline{g}$ represents a rotation/reflection + translation in $\mathbb{E}_d$, and $g$ is the action of the DMS finite group $\mathcal{G}$, forcing a transformation of the internal joint-space configuration. The difference between

---

[5]Technically, the topology of $Q \doteq \mathbb{E}_d \times Q_J$ is referred to as a trivial principal fiber bundle (Ostrowski & Burdick, 1996), with $\mathbb{E}_d$ as the fiber Lie group, and $Q_J$ as the base space. Note that this topology applies to a larger range of dynamical systems than merely floating-base.

[6]We deliberately abuse notation to keep the homogeneous matrix representation $\boldsymbol{X}_B$ of position and orientation, instead of the vector-quaternion representation, common in robotics and computer graphics.

[7]These symmetries are commonly studied since, in conservative systems, translational and rotational symmetries imply the conservation of linear and angular momentum, while time symmetries imply the conservation of energy (Noether, 1918).

$g$ and $\overline{g}$ is highlighted when reformulating eq. (5) for a specific system configuration:

$$\mathcal{L}\left(\begin{bmatrix} \rho_{\mathbb{E}_d}(\overline{g})\boldsymbol{X}_B \\ \hat{\boldsymbol{q}} \end{bmatrix}, \begin{bmatrix} \rho_{\mathbb{E}_d}(\overline{g})\dot{\boldsymbol{X}}_B \\ \dot{\hat{\boldsymbol{q}}} \end{bmatrix}\right) = \mathcal{L}\left(\begin{bmatrix} \rho_{\mathbb{E}_d}(g)\boldsymbol{X}_B \\ \rho_{\mathbb{Q}_J}(g)\ \hat{\boldsymbol{q}} \end{bmatrix}, \begin{bmatrix} \rho_{\mathbb{E}_d}(g)\dot{\boldsymbol{X}}_B \\ \rho_{\mathbb{Q}_J}(g)\ \dot{\hat{\boldsymbol{q}}} \end{bmatrix}\right) \Bigg|_{\rho_{\mathbb{E}_d}(g)\boldsymbol{X} = \rho_{\mathbb{E}_d}(\overline{g})\boldsymbol{X}\rho_{\mathbb{E}_d}(\overline{g})^{\text{-}1}}^{\rho_{\mathbb{E}_d}(\overline{g}) = \pm 1,\ \rho_{\mathbb{E}_d}(g) = 1} \quad (6)$$

What eq. (6) highlights is that with DMS infeasible reflections ($|\rho_{\mathbb{E}_d}(\overline{g})| = -1$) and feasible or infeasible rotations/translations of space are imitated by a feasible ($|\rho_{\mathbb{E}_d}(g)| = 1$) transformation to the system's base and a change in joint-space configuration. Furthermore, the structure of the proper transformation $\rho_{\mathbb{E}_d}(g)\boldsymbol{X} = \rho_{\mathbb{E}_d}(\overline{g})\boldsymbol{X}\rho_{\mathbb{E}_d}(\overline{g})^{\text{-}1}$, along with the properties of the dynamics of symmetrical dynamical systems (eq. (4)), provide a pathway for the identification of $\mathcal{G}$ and the representations $\rho_{\mathbb{Q}}(g) \mid \forall g \in \mathcal{G}$ for any floating-base dynamical system (section 4.3).

## 4.1 DATA AUGMENTATION IN SYSTEMS WITH DMS

Recall from section 3 that point-transformations $g \in \mathcal{G}$ have the same representation $\rho_{\mathbb{Q}}(g)$ for the configuration space Q, its tangent space $\mathrm{T}_{\boldsymbol{q}}\mathrm{Q}$ and any higher order tangent spaces (e.g., the space of generalized accelerations and forces (eq. (3))). Since our floating-base systems' configuration space has the topology $\mathrm{Q} \doteq \mathbb{E}_d \times \mathrm{Q}_J$, this property passes to the representations on $\mathbb{E}_d$ and $\mathrm{Q}_J$. Meaning that the representation $\rho_{\mathbb{E}_d}(g)$ can be used to augment members of $\mathbb{E}_d$ (i.e., points, vectors, and orientations) and members of $\mathbb{E}_d$ higher order tangent spaces (i.e., linear & angular velocities/accelerations). Likewise the representation $\rho_{\mathbb{Q}_J}(g)$ can be used to augment members of $\mathrm{Q}_J$ and its higher order tangent spaces (i.e., joints positions/velocities/accelerations, joint forces/torques).

In practice, this means that any proprioceptive (e.g., joint torques, contact forces) and exteroceptive (e.g., point clouds, terrain height maps, RGBD-images) measurements relevant to the evolution of the system's dynamics, can be augmented solely with combinations of $\rho_{\mathbb{E}_d}(\overline{g})$ and $\rho_{\mathbb{Q}_J}(g)$. Since these measurements live in Q and $\mathbb{E}_d$ and their higher order tangent spaces (see examples in supp.fig 5 and supplementary E.3.1 and E.4.1). To achieve this we need to identify the symmetry group $\mathcal{G}$ and its action representations (section 4.2).

## 4.2 DMS IN THE CASE OF RIGID-BODY DYNAMICS

Until now, we have only assumed our dynamical system is a floating-base system. Now, we assume the system dynamics are ruled by ridig-body dynamics. This means that our system is a collection of $n_B$ interconnected rigid bodies. This is the most frequent scenario in robotics, computer graphics, and experimental biology (see supplementary A). In rigid body dynamics the generalized mass matrix is given by $\mathbf{M}(\boldsymbol{q}) = \sum_k^{n_B} \mathbf{J}_{T_k}(\boldsymbol{q})^{\intercal} m_k \mathbf{J}_{T_k}(\boldsymbol{q}) + \mathbf{J}_{R_k}(\boldsymbol{q})^{\intercal} \mathbf{I}_k \mathbf{J}_{R_k}(\boldsymbol{q})$, being $\mathbf{J}_{T_k}(\boldsymbol{q}) : \mathrm{Q} \to \mathbb{R}^{d \times n}$ and $\mathbf{J}_{R_k}(\boldsymbol{q}) : \mathrm{Q} \to \mathbb{R}^{d \times n}$ the position and orientation Jacobians that are used to map generalized velocities to the linear ($\dot{\mathbf{r}}_k = \mathbf{J}_{T_k}(\boldsymbol{q})\dot{\boldsymbol{q}}$) and angular ($\boldsymbol{w}_k = \mathbf{J}_{R_k}(\boldsymbol{q})\dot{\boldsymbol{q}}$) velocities of the body $k$ (Wieber, 2006). These Jacobians are functions of the kinematic parameters of the system. While $m_k$ and $\mathbf{I}_k$, the mass and inertia of body $k$, represent the dynamic parameters of the system dynamics. A DMS implies symmetries over the kinematic and dynamic parameters of the system, that in practice become useful for the identification of the DMS group $\mathcal{G}$.

**Symmetries of kinematic parameters (Kinematic Tree):** Considering only the kinematic parameters and the equivariance nature of $\mathbf{M}(\boldsymbol{q})$ (eq. (4)), we conclude that a rigid-body system with a symmetry group must have positional and rotational Jacobians that respect $\mathbf{J}_{T_k}(g \cdot \boldsymbol{q}) = \mathbf{J}_{T_k}(\boldsymbol{q})g^{\text{-}1} \wedge \mathbf{J}_{R_k}(g \cdot \boldsymbol{q}) = \mathbf{J}_{R_k}(\boldsymbol{q})g^{\text{-}1} \mid \forall\ g \in \mathcal{G}$. Being $g$ a continuous or discrete symmetry action. In the case of DMS, in which the discrete action $g \in \mathcal{G}$ is designed to imitate the effect of a specific continuous symmetry action $\overline{g} \in \mathbb{E}_d$, we have that the $i^{th}$ body Jacobians should respect:

$$\mathbf{J}_{T_i}(g \cdot \boldsymbol{q})g = \mathbf{J}_{T_k}(\overline{g} \cdot \boldsymbol{q}) = \mathbf{J}_{T_k}(\boldsymbol{q})\overline{g}^{\text{-}1} \wedge \mathbf{J}_{R_i}(g \cdot \boldsymbol{q})g = \mathbf{J}_{R_k}(\overline{g} \cdot \boldsymbol{q}) = \mathbf{J}_{R_k}(\boldsymbol{q})\overline{g}^{\text{-}1} \mid \forall (g, \overline{g}) | g \in \mathcal{G}, \overline{g} \in \mathbb{E}_d \quad (7)$$

Where $k$ is the index of the body of the $\overline{g}$ transformed system (see appendix C.3). In practice, the symmetry in kinematic parameters described in eq. (7) is interpreted as a kinematic tree symmetry (see supplementary C), requiring the discrete action $g$ to result in a kinematic tree indistinguishable from the one obtained by applying the rotation/reflection + translation $\overline{g}$.

**Symmetries of dynamic parameters (Mass and Inertia of rigid-bodies)**: In order for a rigid-body dynamical system to have a DMS the bodies of the system must have symmetric mass distribution or the kinematic tree must be modular (subchains of the tree are symmetric to each other). To understand this morphological constraint consider the base body configuration $\boldsymbol{X}_B \in \mathbb{E}_d$ and the

definition $\rho_{\mathbb{E}_d}(g)\boldsymbol{X}_B = \rho_{\mathbb{E}_d}(\overline{g})\boldsymbol{X}_B\rho_{\mathbb{E}_d}(\overline{g})^{-1}$ in eq. (6). Where the action on the left of $\boldsymbol{X}_B$ is interpreted as a Euclidean transformation of the base in a global reference frame and the action to the right as a transformation in the frame attached to the base.

Recall that, for $g$ and $\overline{g}$ to be Lagrangian-equivalent (eq. (5)) the dynamics of the base body at the configuration $\rho_{\mathbb{E}_d}(g)\boldsymbol{X}_B$ should be identical to the dynamics of the body at $\rho_{\mathbb{E}_d}(\overline{g})\boldsymbol{X}_B$ (eqs. (3) and (5)). Assuming exact kinematic parameter symmetries, both body configurations will have equivalent dynamics if their Inertia matrix $\mathbf{I}_B$ in both configurations is identical . Because in general, $\rho_{\mathbb{E}_d}(g) \neq \rho_{\mathbb{E}_d}(\overline{g})$, the rigid-body Inertia must be invariant to the right transformation $\boldsymbol{X}_B\rho_{\mathbb{E}_d}(\overline{g})^{-1}$. This inertia invariance implies a symmetric mass distribution of the rigid body (see geometric proof in supplementary C.2)). And becomes a key property for the identification of the DMS group $\mathcal{G}$.

As an example consider the robot Solo in fig. 1. This robot is able to imitate two reflections of space $(\overline{g}_t, \overline{g}_s)$ and a $180°$ rotation of space $\overline{g}_r$. This is possible since the base body of the robot has two symmetry planes (see supp.fig 6b), making the inertia of the base $\mathbf{I}_B$, at any arbitrary configuration, invariant under the transformation $\boldsymbol{X}_B\rho_{\mathbb{E}_d}(\overline{g})^{-1} \mid \overline{g} \in \{\overline{g}_t, \overline{g}_s, \overline{g}_r\} \in \mathcal{K}_4$.

**Modular Kinematic Trees**: Theoretically, the previously described constraint of symmetric mass distribution applies to all rigid bodies in the system, limiting the applicability of DMS to diverse floating-base systems. Conveniently, most systems of interest are modular[8], i.e., their kinematic trees are composed of subchains with identical or reflected rigid bodies (e.g., see in supp.fig 5 the identical replication of fingers in the TriFinger robot, or the reflected arms and legs of the humanoid Atlas supp.fig 6a). In such architectures, swapping identical/reflected bodies (and thus subchains of the tree) can satisfy the Inertia invariance required for $\rho_{\mathbb{E}_d}(g)\boldsymbol{X} = \rho_{\mathbb{E}_d}(\overline{g})\boldsymbol{X}\rho_{\mathbb{E}_d}(\overline{g})^{-1}$ without requiring symmetric mass distributions. Refer to appendix C.3 and supp.fig 5 for details and examples.

### 4.3 Identification of DMS group $\mathcal{G}$ in rigid-body dynamics

The identification of the DMS group $\mathcal{G}$ of a floating-base dynamical system, composed of rigid-bodies, can be outlined in four steps (see simple examples in supp.fig 5):

1. Identify the configuration $\boldsymbol{X}_B$ and its associated Inertia $\mathbf{I}_B$. Usually the base body or the CoM.
2. Identify the symmetries in mass distribution as invariances to Euclidean transformations $\overline{g} \in \mathbb{E}_d$ of the reflected $\mathbf{I}_B$. These are the candidate actions that the system could imitate.
3. Identify modularity in the kinematic tree. I.e., all pairs of identical/reflected rigid bodies.
4. From base to end-effectors use eq. (7) to determine for each $\overline{g}$, if the action $g$ and $\rho_{Q_J}(g)$ exists.

## 5 $\mathcal{G}$-Equivariant and $\mathcal{G}$-Invariant Function Approximators

Once we identified the DMS group $\mathcal{G}$ of our system, we know that any proprioceptive or exteroceptive measurements have the same symmetry group $\mathcal{G}$ (section 4.1). Therefore, to improve generalization and sample efficiency, we can exploit the known symmetries of the input $\boldsymbol{x}$ and output $\boldsymbol{y}$ spaces, of any mapping we desire to approximate, by constructing $\mathcal{G}$-equivariant or $\mathcal{G}$-invariant (eq. (1)) NN $f(\boldsymbol{x}; \phi)$, with parameters $\phi$ (Bronstein et al., 2017). This section is built on top of the framework for the construction of $\mathcal{G}$-equivariant NN of Finzi et al. (2021b). Where our main motivation is to address the limitations that prohibit the construction of large-scale $\mathcal{G}$-equivariant NN (see supplementary D), which are ubiquitous in real-life applications.

Consider $f(\boldsymbol{x}; \phi)$ to be composed of multiple perceptron (or convolutional) layers of the form $^l\boldsymbol{y} := \sigma(^l\boldsymbol{W}^l\boldsymbol{x} + {}^l\boldsymbol{b})$, where $^l\boldsymbol{x} \in \mathbb{R}^n$, $^l\boldsymbol{y} \in \mathbb{R}^m$, $^l\boldsymbol{W} \in \mathbb{R}^{m \times n}$ and $^l\boldsymbol{b}$ are the $l$ layer's linear map and bias, respectively; and $\sigma : \mathbb{R} \to \mathbb{R}$ is a strictly monotonic nonlinearity (Ravanbakhsh et al., 2017). With this parametrization, the equivariance constraints of eq. (1) can be reduced to constraints on the linear map $\boldsymbol{W}$ (dropping the layer index $l$ for notation clarity): [9]

$$\rho_{out}(g)\boldsymbol{W} = \boldsymbol{W}\rho_{in}(g) \mid \forall\, g \in \mathcal{G} \quad \Longleftrightarrow \quad (\rho_{\boldsymbol{W}}(g) - \boldsymbol{I})\boldsymbol{w} = \boldsymbol{0} \mid \forall\, g \in \mathcal{G}. \tag{8}$$

The RHS of eq. (8) is a reformulation of the equivariance constraints as a standard set of linear equations, defining $\rho_{\boldsymbol{W}}(g) = \rho_{out}(g) \otimes \rho_{in}(g^{-1})^{\top} \in \mathbb{R}^{mn \times mn}$ as the representation of the group

---

[8]This covers most flying/swimming/legged robots, animals and animated characters (supplementary A)

[9]A similar analysis can be made for the bias vector $\boldsymbol{b}$.

action acting on the linear map as a result of a semi-direct product[10] of the input and output group actions ($\otimes$ stands for the Kronecker product) and $\boldsymbol{w} = vec(\boldsymbol{W}) \in \mathbb{R}^{mn}$ as a vectorized version of $\boldsymbol{W}$ (refer to Finzi et al. (2021b) for details). Since the constraint imposed by each $g$ is linear in $\boldsymbol{W}$, we can stack them into a single system of linear equations $\boldsymbol{Cw} = \boldsymbol{0}$. The nullspace of this system of equations $\boldsymbol{B} \in \mathbb{R}^{mn \times r}$ describes the $r$ basis vectors spawning the entire space of equivariant linear maps. Allowing to parameterize all $\mathcal{G}$-equivariant $\boldsymbol{W}$ as:

$$\boldsymbol{w} = \sum_k^r c_k \boldsymbol{B}_{:,k} \quad \Longleftrightarrow \quad \boldsymbol{W} = \sum_k^r c_k \text{ unvec}(\boldsymbol{B}_{:,k}) \doteq \sum_k^r c_k \mathbf{B}_{:,:,k}. \tag{9}$$

Where the basis coefficients $\boldsymbol{c} \in \mathbb{R}^r$ represent the free variables of the system of equations and the trainable parameters of the equivariant layer (see fig. 1 right).

**Dealing with memory complexity of equivariant layers:** An equivariant layer needs to store the matrices $\rho_{\boldsymbol{w}}(g)$ and $\boldsymbol{B}$, in addition to the typical memory complexity of a perceptron or convolutional layer. These matrices' memory complexity quickly becomes intractable for moderate input-output dimensions (see supp.table 1). Fortunately, finite symmetry groups have sparse action matrix representations, resulting in both of the aforementioned matrices being sparse. Our implementation[2] extends the Pytorch API from Finzi et al. (2021b) to process finite groups with sparse matrix definitions limiting the additional memory footprint of equivariant layers to a minimum.

**Dealing with the computational complexity of determining the equivariant basis $B$:** Computing $\mathbf{B}$ for a layer is a process with high computational complexity. Finzi et al. (2021b) proposes a Krylov gradient-based method able to handle both finite and Lie groups' arbitrary regular representations. While Van der Pol et al. (2020) approximates $\boldsymbol{B}$ through SVD of a matrix $\bar{\boldsymbol{W}} \in \mathbb{R}^{z \times mn}$ ($z \geq mn$). Both approaches run in polynomial time $\mathcal{O}(r^2(mn)^2)$ (prohibiting their use in large dimensional spaces) and approximate the space rank $r$ numerically.

Fortunately, DMS groups $\mathcal{G}$ are finite and have, in general, generalized permutation matrices as regular representations. Enabling the computation of $\boldsymbol{B}$ in linear time (see supplementary B): Note that the constraints imposed by each $\rho_{\boldsymbol{w}}(g)$ result in parameter sharing constraints (e.g., $w_{10} = -w_2 = \ldots = w_0$). In these cases, every vector of the null-space of $\boldsymbol{C}$ (i.e., $\boldsymbol{B}_i$) simply describes the sharing scheme of a free variable of the system of equations (i.e., the trainable parameter $c_i$), and this sharing scheme is nothing else but one of the unique $r$ orbits of the dimensions of $\boldsymbol{w}$ when transformed by all group actions, e.g., $\mathcal{G} \cdot w_{10} = \{g \cdot w_{10} : \forall g \in \mathcal{G}\} = \{w_{10}, -w_2, \ldots, w_0\}$ (see the parameter orbits of length 4 in fig. 1-right, for $\mathcal{K}_4$). The orbits of all $w \in \boldsymbol{w}$ are trivially computed with $[\boldsymbol{w}, \rho_{\boldsymbol{w}}(g_1)\boldsymbol{w}, \ldots, \rho_{\boldsymbol{w}}(g_{|\mathcal{G}|})\boldsymbol{w}]$, while the unique $r$ orbits can be identified in $\mathcal{O}(mn)$ time. Our proposed solution can be thought of as a linear-time version of Ravanbakhsh et al. (2017).

**Optimal trainable parameter initialization for equivariant layers:** Proper initialization of the equivariant layer's trainable parameters $\boldsymbol{c}_l$ (eq. (9)) is required to avoid activations/gradients from vanishing or exploding (Klambauer et al., 2017). Following the same derivation of the Kaiming initialization (He et al., 2015) (see supplementary D.2), we can conclude that the parameters should be initially sampled from a distribution with $\text{Var}(\boldsymbol{c}_l) = {}^m/\lambda_{\mathbf{B}}\gamma_\sigma$, to ensure constant variance of activations throughout the network layers (see supp.fig 8). Where $\lambda_{\mathbf{B}} = \sum_i^m \sum_j^n \sum_k^r \mathbf{B}_{i:j:k}^2$ and $\gamma_\sigma$ is a nonlinearity dependant scalar (e.g., $\gamma_{\text{ReLu}} = {}^1/2$, $\gamma_{\text{SeLu}} = 1$ following Klambauer et al. (2017)). This initialization depends only on $\mathbf{B}$. Thus, is applicable to any Lie or finite group.

**Reduction of trainable parameters in equivariant layers:** Determining analytically the number of trainable parameters (i.e. the rank $r$) of an $\mathcal{G}$-equivariant layer is, in general, an unresolved problem. However, for DMS groups, we show on supplementary D.1 that the number of trainable parameters of a $\mathcal{G}$-equivariant layer can range from ${}^{|\boldsymbol{w}|}/|\mathcal{G}| \leq r \leq |\boldsymbol{w}|$, depending on the number of dimensions of the input-output spaces left invariant by the symmetry actions. In practice, this implies that for a $\mathcal{G}$-equivariant layer without any input-output fixed points (e.g., all intermediate layers of a $\mathcal{G}$-equivariant NN), the number of trainable parameters is reduced by ${}^1/|\mathcal{G}|$ being $|\mathcal{G}|$ the group order. Therefore a $\mathcal{G}$-equivariant architecture with $\mathcal{G} = \mathcal{C}_2$ (supp.fig 6a) (or $\mathcal{G} = \mathcal{K}_4$, see fig. 1) will have approximately ${}^1/2$ (or ${}^1/4$) of the trainable parameters of an unconstrained NN of the same architectural size. The reduction of parameters is caused by the parameter sharing constraints (eq. (9)) and is visually depicted in fig. 1-right.

---

[10] Note that Finzi et al. (2021b) considers always a direct product of the input-output symmetry actions. However, for DMS as the input and output groups are isomorphic, a direct product over-contains the models to symmetries not present in the data.

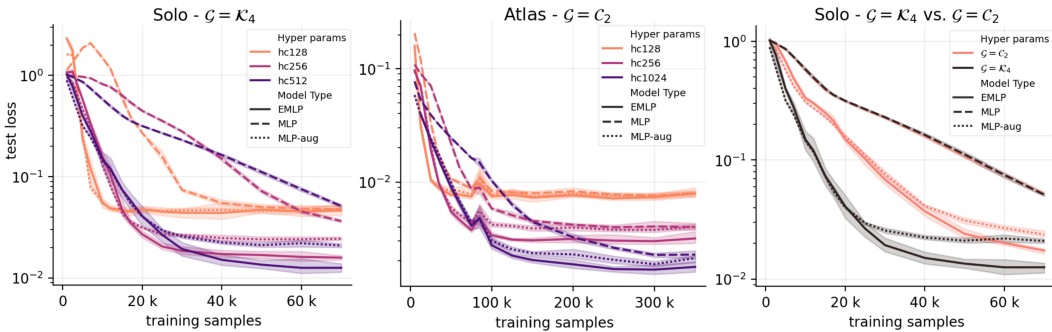

Figure 2: **CoM-estimation results comparing MLP, MLP-aug, and EMLP models**. **Left and Middle:** Test set sample efficiency, for robot Solo and Atlas, of model variants with different capacities (number of hidden layers' neurons $hc$). **Right:** Sample efficiency for robot Solo (fig. 1), of models with $hc = 512$, when exploiting $\mathcal{G} = \mathcal{K}_4$ (sagittal and traversal symmetries) and $\mathcal{G} = \mathcal{C}_2 = \{e, g_s\} \subset \mathcal{K}_4$ (only sagittal symmetry). Reported values represent the average and standard deviation across 10 different seeds.

## 6    EXPERIMENTS

We present two experiments of supervised learning, a regression application using synthetic data and a classification application using real-world data. Both experiments aim to illustrate the versatility of DMSs for data augmentation and training of equivariant functions, along with the impact on the model's sample efficiency and generalization capacity when exploiting DMSs. While we keep the presentation concise, all the technical aspects are detailed in supplementary E and [2].

**CoM Momentum Estimation (Regression)**: In this experiment, we train a NN to approximate a robot's center-of-mass momentum given the joint-space position and velocities: $\boldsymbol{h} = \mathbf{A}_G(\hat{\boldsymbol{q}})\dot{\hat{\boldsymbol{q}}}$. Where $\boldsymbol{h} = [\boldsymbol{l}^\intercal \ \mathbf{k}^\intercal]^\intercal$ are the linear $\boldsymbol{l}$ and angular $\mathbf{k}$ momentum components And $\mathbf{A}_G$ is the Centroidal Momentum Matrix (CMM) of Orin et al. (2013). This analytical function is highly non-linear and $\mathcal{G}$-equivariant to the robot's DMS group $\mathcal{G}$. Consequently, the function approximator is expected to be equivariant or to approximate equivariance.

We test two robots: (1) Atlas, a $n_J = 30[\text{DoF}]$ humanoid robot with a reflection DMS group $\mathcal{G} = \mathcal{C}_2$ (see supp.fig 6a). (2) Solo, a $n_J = 12[\text{DoF}]$ quadruped robot with the Klein-4 group as DMS group $\mathcal{G} = \mathcal{K}_4$ (see fig. 1-left). At the same time, we compare three variants of a function approximation: a standard Multi-Layer Perceptron (MLP), a version of the MLP trained using data-augmentation (MLP-aug), and a version of the MLP with hard-equivariance constraints (E-MLP).

On fig. 2-left-&-middle, we compare the model variants. For both robots and all model capacities, the E-MLP and MLP-Aug outperform MLP on sample efficiency (better generalization with less data) and robustness to overfitting when training data is scarce. Comparing the E-MLP and MLP-Aug model variants, we see that the lower capacity versions behave similarly, but as capacity increases, E-MLP starts to show better sample efficiency and generalization. Lastly, on (fig. 2-right) we compare, for the robot Solo, the performance of the model variants when exploiting the robots' entire symmetry group ($\mathcal{K}_4$) and a subgroup of the real symmetry group ($\mathcal{C}_2 \subset \mathcal{K}_4$). The results indicate that sample efficiency and generalization capacity increase with the number of true symmetries of the data exploited.

**Static-Friction-Regime Contact Detection (Classification)**: This experiment uses the dataset presented in Lin et al. (2021) for the estimation of static-friction-regime contacts in each of the four legs of the Mini-Cheetah quadruped robot. The dataset samples, collected in the real-world, consist of the history of the past 150 time-frames of proprioceptive data collected from inboard sensors of the robot during locomotion with various gaits and over several terrains, $\boldsymbol{x} \in \mathbb{R}^{54 \times 150}$, and the ground truth contact state of the robot $\boldsymbol{y} \in \mathbb{R}^{16}$, estimated off-line using a non-causal algorithm (i.e., dependant on the past and future). The objective is to train a causal function approximator $f(\boldsymbol{x}; \boldsymbol{\phi})$ for estimating the contact state.

The real-world Mini-Cheetah robot has an approximate reflection DMS $\mathcal{G} \approx \mathcal{C}_2$. Hence, both the proprioceptive data $\boldsymbol{x}$ and the contact state $\boldsymbol{y}$ share the symmetry group $\mathcal{G}$ (see supplementary E.4). We compare three variants of function approximators: the original Conv-NN architecture of Lin

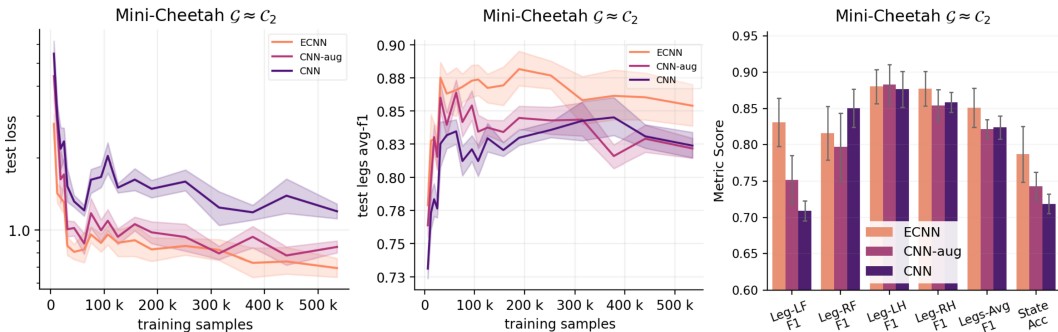

Figure 3: **Static-Friction-Regime contact detection results comparing CNN, CNN-aug, and ECNN. Left:** Sample efficiency in log-log scale. **Middle:** Average legs F1-score vs. training samples. **Right:** Classification metrics on test set performance of models trained with the entire training set. Selected metrics are contact-state ($y \in \mathbb{R}^{16}$) accuracy (Acc) and f1-score (F1) for each leg binary contact state. Due to the sagittal symmetry of the robot the left front (LF) and right front (RF) legs are expected to be symmetric, as the left hind (LH) and right hind (RH) legs. F1-score is presented considering the dataset class imbalance (see supplementary E.4 and supp.fig 7). Reported values represent the average and standard deviation across 8 different seeds.

et al. (2021) (CNN), a version of CNN trained using data-augmentation (CNN-aug), and a version of CNN with hard-equivariance constraints (E-CNN).

The sample efficiency of the model variants and the average legs contact state classification are illustrated in fig. 3-left-&-middle. Where the equivariant model E-CNN presents better generalization, performance, and robustness to dataset biases (see supp.fig 7) than the unconstrained models across all training set sizes, followed by CNN-aug. In fig. 3-right we evaluate test set classification metrics when using the entire training data. The E-CNN model outperforms both CNN-aug and CNN on contact state classification and average leg contact detection. Of relevant importance is the mitigation of suboptimal asymmetries of the models by exploiting symmetries. Preventing the model to favor the classification of one leg above others (fig. 3). Refer to appendix E.5 for details.

# 7 CONCLUSIONS & DISCUSSION

In this work, we present the concept of Discrete Morphological Symmetry (DMS). These are discrete symmetries of dynamical systems evolving in Euclidean space, that are associated with the capability of the system to imitate Euclidean transformations (rotations/reflections and translations) with discrete changes in the system's internal state configuration. With this formalism, we can describe the bilateral and radial symmetries that are ubiquitous in robotic systems and animals in nature. By studying these symmetries with the language of Group Theory, we propose a mechanism for the identification of the finite DMS group $\mathcal{G}$, and of the representations of the symmetry actions in the system's state variables and relevant proprioceptive and exteroceptive measurements.

Having made the connection between Dynamical Systems and Group Theory, we show why and how these symmetries should be exploited in data-driven applications—to obtain improvements in sample efficiency and generalization capacity—, either by using data-augmentation or $\mathcal{G}$-equivariant Neural Networks. For the latter, we present practical contributions addressing the implementation drawbacks (intractable computational and memory complexity) of using $\mathcal{G}$-equivariant architectures for real-world applications. Additionally, we release open-access code enabling the rapid prototyping of $\mathcal{G}$-equivariant Neural Networks for the exploitation of DMS in applications processing data from rigid-body dynamics (e.g., robotics, computer graphics, and computational biology).

Lastly, we present empirical results supporting our claims on two data-driven applications using synthetic and real-world data from three different robots. In both experiments improvements in sample efficiency and generalization are obtained by exploiting the morphological symmetries bias, motivating the use of this technique in applications using data from simulation and/or the real world.

**Limitations:** For a detail account of limitations see supplementary B.

REPRODUCIBILITY STATEMENT

The experimental setup used in the experiments is described in section 6 and supplementary E. Moreover, our implementation[2] will be open-access, where any interested party can find: (1) The original scripts used to run the experiments and generate the results, (2) the parameters of the models used for comparison in the experiments, avoiding the need to retrain the models to test the results, (3) the datasets used in each of our experiments, including the custom partitioning of the dataset of Lin et al. (2021), (4) the scripts used to summarize the results into the figures used in this paper, and (5) 3D interactive environments that allow for the visualization of the morphological symmetries, one of this environments was used for capturing fig. 1-left and its 3D animation.

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

# SUPPLEMENTARY
## ON DISCRETE SYMMETRIES OF ROBOTICS SYSTEMS:
## A GROUP-THEORETIC AND DATA-DRIVEN ANALYSIS

## A  APPLICATIONS OF DISCRETE MORPHOLOGICAL SYMMETRIES

In this section, we provide our perspective on possible applications where DMS can be of value and the respective fields of knowledge where these applications play an important role.

All applications of DMS *for data-driven techniques* fall within two categories (1) data augmentation of proprioceptive and exteroceptive measurements, and (2) $\mathcal{G}$-Equivariant function approximation. The fields of knowledge that can benefit from the aforementioned applications are:

- **Biology, Biomechanics, and Experimental Veterinary**. Studying the biomechanics and dynamics of animals in nature is becoming a fundamental area of the fields of Biology, Biomechanics, and Experimental Veterinary (Wei & Kording, 2018). Considering that around $99\%$ of eumetazoans (most species excluding sponges and other sea species) are Bilaterian (Ferretti et al., 2020) (i.e., having approximate $\mathcal{C}_2$ symmetry) or Radiatal (i.e., having approximate $\mathcal{C}_n \mid n \geq 2$ symmetry), DMS become a flexible and natural approach to study the data gathered from the study of these systems, especially of vertebrates, whose dynamics are often approximated to rigid body dynamics (Wu et al., 2022).

  The process of study of animal motion dynamics normally involves the use of motion capture data of animal motions using marker-based (Prankel et al., 2016) or marker-less (Mathis et al., 2018) sensor pipelines. The data of the markers is then either directly processed or fitted to kinematic models (that make the assumption of exact kinematic symmetries) and then processed for information retrieval. DMS offers a clear approach to mitigate the cost of data collection by providing a simple approach for data augmentation, and for the construction of $\mathcal{G}$-Equivariant NN to process the dynamics of the kinematic chains.

- **Computer Graphics & Vision**. Computer Graphics is perhaps the de facto application field of exact DMS. In this area, the kinematic structure of animated characters is assumed to be symmetric, as they often model the behavior of living vertebrates with $\mathcal{C}_n \mid n \geq 2$ symmetry. The recorded trajectories are obtained through motion capture data or expert artist animations, and to the author's knowledge, the trajectories are seldomly augmented to their symmetric equivalents. In this field, NNs are commonly used to learn projection spaces where motion matching and animation interpolation becomes an easier problem than in minimal coordinate space (Holden et al., 2015; Starke et al., 2022) or control policies for physics-based animation (Peng et al., 2018; Ma et al., 2021). However, the exploitation of the DMS inductive bias is not common in the field and has been approached (with specialized and costly algorithms) solely for the $\mathcal{C}_2$ symmetry in Yeh et al. (2019); Ordonez-Apraez et al. (2022); Abdolhosseini et al. (2019); Wu et al. (2022).

- **Robotics**. NN function approximators are becoming a valuable tool in robotics applications of perception, control (Miki et al., 2022) and state-estimation (Lin et al., 2021). In all applications, NNs are used to approximate functions processing proprioceptive or exteroceptive measurements related to the evolution of the dynamics of the robot. Despite the majority of legged platforms (Radford et al., 2015; Grimminger et al., 2020; Miki et al., 2022) and manipulators having $\mathcal{C}_n \mid n \geq 2$ symmetries, the use of data augmentation (or $\mathcal{G}$-equivariant NN) to mitigate the high cost and risks of collecting data with robotic systems in the real-world (or simulation) is not a widespread technique. We believe the framework of DMS and our open-access code can contribute to the adoption of $\mathcal{G}$-equivariant function approximators in the field.

- **Control** In the case of model-based control, it is common to exploit the symmetries of the Euclidean Lie Group in which the robot evolves (section 3)(Wu & Sreenath, 2015) and to inherently exploit the equivariance of inertial forces (eq. (4)) by assuming approximate or exact symmetries in the kinematic and dynamic parameters of the dynamics model (Mastalli et al., 2022). However, to the authors' knowledge, no approach exploits DMSs (and especially the discrete symmetries of the joint space $Q_J$) in applications of exploration

of space, planning, and trajectory optimization, where DMS offers a technique to avoid the computation of symmetric trajectories.

In model-free control, specifically in reinforcement learning (RL), the exploitation of symmetries in the dynamics implies mitigation of the sample inefficiency and sensibility to local optima that these learning algorithms have. Previous works have shown the impact of symmetry data-augmentation (Weissenbacher et al., 2022; Ordonez-Apraez et al., 2022) and of $\mathcal{G}$-equivariant function approximators (Van der Pol et al., 2020; Ordonez-Apraez et al., 2022) on model-free RL.

## B  LIMITATIONS

Our work makes two main assumptions:

1. **Symmetries are exact**: By assuming that a dynamical system has exact and not approximate symmetries we are departing from the real-world nature of DMSs, since for any robotic system in the real-world the manufacturing and assembly process introduces errors/tolerances in the kinematic and dynamic parameters of each of the robot's bodies. Likewise, the dynamics of animals in nature are not perfectly equivariant since morphological symmetries are only approximate symmetries. Although exact symmetries seem to be a strong assumption, in practice, the reality is that it is a common assumption in the fields of robotics and control theory, in which idealized models of the dynamics are often assumed, implying exact DMSs through the exact symmetries in kinematic and dynamic parameters (which are responsible for the equivariant nature of the generalized mass matrix $\mathbf{M}(g \cdot \boldsymbol{q}) = g\mathbf{M}(\boldsymbol{q})g^{-1}$ (eq. (4)) and therefore, for the equivariance of inertial, centripetal, gravitational, and Coriolis generalized forces).

   On section 6 we show that the exact symmetry bias is justifiable and beneficial for learning function approximators processing the dynamics of approximately symmetrical systems in the real world. However, the authors highlight the necessity to properly address the case of approximate equivariance, which we leave to future work. To address this case, system identification techniques Simpkins (2012) have been wildly used to approximate the deviation of the kinematic and dynamic parameters from the assumed values. While in the case of $\mathcal{G}$-equivariant NN Wang et al. (2022); Finzi et al. (2021a) provide clear and valuable approaches to learn approximate $\mathcal{G}$-equivariant NN.

   It is relevant to highlight that, in physics-based simulation, the most common practice is to work with the idealized model of dynamics. Thus, the assumption of exact symmetries is justifiable and encouraged in applications where simulation is a relevant tool (see supplementary A).

2. **Linear time computation of the basis $\mathbf{B}$ of equivariant linear maps is restricted to group actions with regular matrix representations that are generalized permutation matrices**: The algorithm for computing $\mathbf{B}$ in $\mathcal{O}(mn)$ time and determining analytically the rank $r$ of this space (see section 5) is restricted to the scenario where $\rho_{\boldsymbol{w}}(g) \in \mathbb{R}^{nm \times nm}$ is a generalized permutation matrix (which occurs when both $\rho_{in}(g)$ and $\rho_{out}(g)$ are also generalized permutation matrices). Although this assumption always holds true for $\mathcal{G} = \mathcal{C}_2$ (the most common DMS) and for all action representations described in section 6 and supp.fig 5 and fig. 1, in general, it might not hold for $|\mathcal{G}| > 2$. In cases where either $\rho_{in}(g)$ and $\rho_{out}(g)$ are not generalized permutation matrices, the computation of $\mathbf{B}$ can be approached using the Krylov subspace method proposed by Finzi et al. (2021b) with complexity $\mathcal{O}(r^2(mn)^2)$ and numerical approximation of $r = rank(\mathbf{B})$.

   Although this seems like a strong assumption, consider that in the case of DMS groups $\mathcal{G}$:

   - All action representations $\rho_{\mathrm{Q}_J}(\cdot)$, acting on the joint-space manifold $\mathrm{Q}_J$ and its tangent space, are generalized permutation matrices. This property holds true when using the common convention of minimum coordinates for $\boldsymbol{q}$ and $\hat{\boldsymbol{q}}$, in which each vector of the orthogonal basis of $\mathrm{Q}_J$ corresponds with a degree of freedom of the system. With this convention any DMS action $g$ acting on a single degree of freedom can be defined as a function of a single degree of freedom $g \cdot \hat{q}_i = g(\hat{q}_j) \quad s.t. \quad \hat{q}_i, \hat{q}_j \in \hat{\boldsymbol{q}}$.
   - All action representations $(\rho_{in}(g), \rho_{out}(g))$ of the latent vector spaces of internal layers of an equivariant neural network (e.g., $^l\boldsymbol{z}$ in fig. 1) can be arbitrarily parameterized

(while respecting the group axioms). Singe $\mathcal{G}$ is by definition a finite symmetry group, we can parameterize $\rho_{in}(g)$ and $\rho_{out}(g)$ to be generalized permutation matrices.

## C   PROPERTIES OF ROBOTIC SYSTEMS WITH MORPHOLOGICAL SYMMETRIES

For clarity of the explanation, let us imagine two different Euclidean spaces and two versions of the robot: the original space (with reference frame $o$) and robot with coordinates $\boldsymbol{q}$ and $\dot{\boldsymbol{q}}$, and the virtual rotated/reflected space (with a reference frame $\overline{o}$, with configuration ${}^{o}\boldsymbol{X}_{\overline{o}} = \begin{bmatrix} \boldsymbol{R}_{\overline{g}} & \mathbf{r}_{\overline{o}} \\ \mathbf{0} & 1 \end{bmatrix}$) and virtual robot with coordinates $\overline{g} \cdot \boldsymbol{q}$ and $\overline{g} \cdot \dot{\boldsymbol{q}}$ referenced to $\overline{o}$. Noting that in the case of a reflection, the virtual robot has reflected versions of each rigid body.

For eqs. (3) and (5) to hold, there must exist an action $g \in \mathcal{G}$ transforming the real robot configuration $g \cdot \boldsymbol{q}, g \cdot \dot{\boldsymbol{q}}$ resulting in the same kinetic energy as the virtual robot's kinetic energy:

$$\mathcal{T}(g \cdot \boldsymbol{q}, g \cdot \dot{\boldsymbol{q}}) = \tfrac{1}{2}\sum_{i=1}^{n_B} m_i \dot{\mathbf{r}}_{g,i}^{\mathsf{T}} \dot{\mathbf{r}}_{g,i} + \boldsymbol{w}_{g,i}^{\mathsf{T}} \mathbf{I}_i \boldsymbol{w}_{g,i} \doteq \tfrac{1}{2}\sum_{k=1}^{n_B} \overline{m}_k \dot{\overline{\mathbf{r}}}_k^{\mathsf{T}} \dot{\overline{\mathbf{r}}}_k + \overline{\boldsymbol{w}}_k^{\mathsf{T}} \overline{\mathbf{I}}_k \overline{\boldsymbol{w}}_k = \mathcal{T}(\overline{g} \cdot \boldsymbol{q}, \overline{g} \cdot \dot{\boldsymbol{q}}),$$
(10)

where $\dot{\mathbf{r}}_{g,i}$, $\boldsymbol{w}_{g,i}$, $m_i$ and $\mathbf{I}_i$ are the linear and angular velocity, mass, and inertia matrix of the transformed body $i$ (referenced to $o$). Likewise, $\dot{\overline{\mathbf{r}}}_i$, $\overline{\boldsymbol{w}}_i$, $\overline{m}_i$ and $\overline{\mathbf{I}}_i$ are the equivalent quantities for the virtual robot and body $i$ (referenced to $\overline{o}$).

### C.1   KINEMATIC SYMMETRIES:

Ignore momentarily the influence of the mass and inertia in terms of the real and virtual bodies. We can assert that for supp.eq 10 to hold, the transformed configuration should result in a kinematic tree indistinguishable from the virtual robot's. Thus, for everybody $i$ in the real robot kinematic tree, there should exist an equivalent virtual body $k$ (as seen in supp.fig 5, not always $k = i$). By equating the linear and angular velocities of the real and virtual bodies, referenced to $o$, and expressing the velocities as functions of the generalized coordinates we obtain:

$$\begin{aligned} \dot{\mathbf{r}}_{g,i} = \dot{\overline{\mathbf{r}}}_k &\doteq \boldsymbol{R}_{\overline{g}} \cdot \dot{\mathbf{r}}_k & \boldsymbol{w}_{g,i} = \overline{\boldsymbol{w}}_k &\doteq |\boldsymbol{R}_{\overline{g}}|\boldsymbol{R}_{\overline{g}} \cdot \boldsymbol{w}_k \\ \mathbf{J}_{T_i}(g \cdot \boldsymbol{q})g \cdot \dot{\boldsymbol{q}} = \boldsymbol{R}_{\overline{g}} \cdot \mathbf{J}_{T_k}(\boldsymbol{q})\dot{\boldsymbol{q}} \quad & \mathbf{J}_{R_i}(g \cdot \boldsymbol{q})g \cdot \dot{\boldsymbol{q}} = |\boldsymbol{R}_{\overline{g}}|\boldsymbol{R}_{\overline{g}} \cdot \mathbf{J}_{R_k}(\boldsymbol{q})\dot{\boldsymbol{q}}, \\ \mathbf{J}_{T_i}(g \cdot \boldsymbol{q})g = \boldsymbol{R}_{\overline{g}} \cdot \mathbf{J}_{T_k}(\boldsymbol{q}) \quad & \mathbf{J}_{R_i}(g \cdot \boldsymbol{q})g = |\boldsymbol{R}_{\overline{g}}|\boldsymbol{R}_{\overline{g}} \cdot \mathbf{J}_{R_k}(\boldsymbol{q}), \end{aligned}$$
(11)

where $\mathbf{J}_{T_i}(\boldsymbol{q})$, $\mathbf{J}_{R_i}(\boldsymbol{q}) \in \mathbb{R}^{3 \times n}$ are the position and orientation analytical Jacobians (describing the instantaneous velocity vectors contributed by each DoF to body $i$) of the real robot at a configuration $\boldsymbol{q}$ (Wieber, 2006). Formulating supp.eq 11 for each of the $n_B$ bodies of the robot we obtain at best $n_B \times 3 \times n$ non-linear equations that can be used to assert if $g$ exists. In practice, the action representation $\rho_{\mathbb{Q}}(g)$ and especially its component acting on the joint space $\rho_{\mathbb{Q}_J}(g)$ can be trivially determined by solving supp.eq 11 (or equivalently eq. (7)) for each body from top to bottom of the kinematic tree (i.e., base first, end-effectors last), if $g$ exists.

### C.2   REFLECTIONS/ROTATIONS REQUIRE MODULARITY OR SYMMETRIC RIGID BODIES

Let us assume kinematic symmetry and direct our attention now to the influence of the mass and inertia terms on the kinetic energy of a single rigid body when it is transformed with the action $g$, which imitates a true reflection of space $\overline{g}$. Focus on the first two columns of supp.fig 4. Because of the kinematic symmetry the CoM of the reflected and transformed bodies coincide, both bodies have equivalent linear components of kinetic energy. However, for arbitrary rigid bodies, the reflected and transform bodies will have different angular components of kinetic energy. Note that in the general case, the transformed and reflected bodies' inertia will differ, thus even if both bodies have the same angular velocities, their kinetic energy will differ.

Let $p$, $p_{\overline{g}}$ and $p_g$ be frames located at the CoM of the original, reflected and transformed bodies, aligned with the principal axes of inertia of each of the bodies. Similarly, denote ${}^{o}\mathbf{I}$ and ${}^{o}\mathbf{I}_g$ as the original and transformed bodies inertias referenced to $o$, and ${}^{\overline{o}}\overline{\mathbf{I}}_{\overline{g}}$ as the reflected body inertia referenced to the reflected Euclidean space $\overline{o}$. In order to comply with supp.eq 11, we must ensure

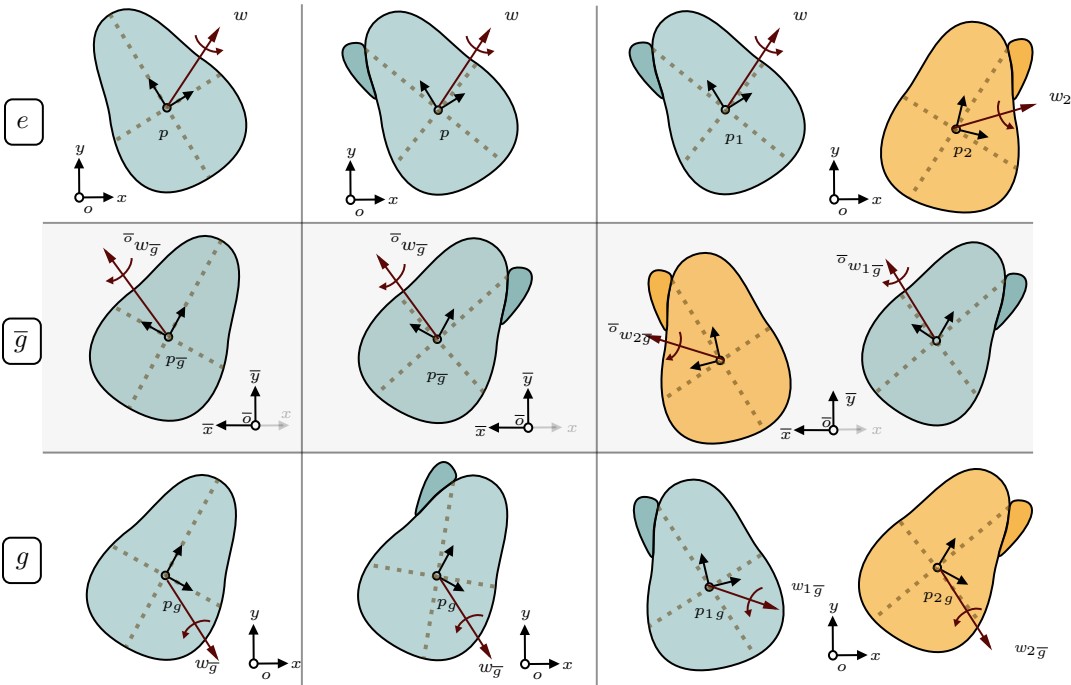

Supplementary Figure 4: Properties of bodies capable of imitating a true reflection $\overline{g}$ of space (w.r.t the $yz$-plane in this case), with a proper transformation $g$ involving only a rotation and translation. The first row shows the original bodies with their respective angular velocities $\boldsymbol{w}$, subjected to trivial symmetry transformation $e$ (dashed lines represent the principle axes of inertia of the bodies), and the second and third rows display the effect of $\overline{g}$ and $g$ on the bodies and angular velocities, respectively. The first column displays a rigid body with symmetric mass distribution, for which $g$ exists, as the reflected and rotated bodies share an equivalent angular kinetic energy. The second column shows a rigid body with asymmetrical mass distribution, for which the rotation $g$, that produces a kinematic symmetry, results in the reflected and rotated bodies having different angular kinetic energies (eq. (2)). The third column shows two bodies with asymmetrical mass distributions, each a reflected version of the other, in this case, the action $g$ swaps bodies configurations to imitate the configuration and energy state of the reflected bodies transformed with $\overline{g}$. Angular velocity is a pseudo-vector (or axial-vector), for which a reflection transformation is computed as $\boldsymbol{w}_{\overline{g}} = |\boldsymbol{R}_{\overline{g}}|\boldsymbol{R}_{\overline{g}}\boldsymbol{w}$ (see Quigley (1973)).

that:

$$
{}^{o}\boldsymbol{w}_{\overline{g}}^{\mathsf{T}}{}^{o}\mathbf{I}_{g}{}^{o}\boldsymbol{w}_{\overline{g}} = {}^{\overline{o}}\boldsymbol{w}_{\overline{g}}^{\mathsf{T}}{}^{\overline{o}}\overline{\mathbf{I}}_{\overline{g}}{}^{\overline{o}}\boldsymbol{w}_{\overline{g}},\,,
$$

$$
(\boldsymbol{R}_{\overline{g}}{}^{o}\boldsymbol{w})^{\mathsf{T}}{}^{o}\mathbf{I}_{g}\,(\boldsymbol{R}_{\overline{g}}{}^{o}\boldsymbol{w}) = {}^{\overline{o}}\boldsymbol{w}_{\overline{g}}^{\mathsf{T}}{}^{\overline{o}}\overline{\mathbf{I}}_{\overline{g}}{}^{\overline{o}}\boldsymbol{w}_{\overline{g}} \quad | \; {}^{o}\boldsymbol{w}_{\overline{g}} = |\boldsymbol{R}_{\overline{g}}|\boldsymbol{R}_{\overline{g}}{}^{o}\boldsymbol{w}\,,
$$

$$
{}^{o}\mathbf{I}_{g} = \boldsymbol{R}_{\overline{g}}{}^{\overline{o}}\overline{\mathbf{I}}_{\overline{g}}\boldsymbol{R}_{\overline{g}} \quad | \; {}^{o}\boldsymbol{w} \equiv {}^{\overline{o}}\boldsymbol{w}_{\overline{g}},\quad \boldsymbol{R}_{\overline{g}}\boldsymbol{R}_{\overline{g}} = \boldsymbol{I},
$$

$$
{}^{o}\boldsymbol{R}^{p_g}{}^{p_g}\mathbf{I}_{g}{}^{o}\boldsymbol{R}^{p_g\,\mathsf{T}} = \boldsymbol{R}_{\overline{g}}{}^{\overline{o}}\boldsymbol{R}^{p_{\overline{g}}}{}^{p_{\overline{g}}}\overline{\mathbf{I}}_{\overline{g}}{}^{\overline{o}}\boldsymbol{R}^{p_{\overline{g}}\,\mathsf{T}}\boldsymbol{R}_{\overline{g}} \quad | \; {}^{a}\mathbf{I} = {}^{b}\boldsymbol{R}^{a\,b}\mathbf{I}^{b}\boldsymbol{R}^{a\,\mathsf{T}},
$$

$$
{}^{o}\boldsymbol{R}^{p_g}{}^{p_g}\mathbf{I}_{g}{}^{o}\boldsymbol{R}^{p_g\,\mathsf{T}} = \boldsymbol{R}_{\overline{g}}{}^{o}\boldsymbol{R}^{p}{}^{p}\boldsymbol{R}^{p_{\overline{g}}}{}^{p_{\overline{g}}}\overline{\mathbf{I}}_{\overline{g}}{}^{p}\boldsymbol{R}^{p_{\overline{g}}\,\mathsf{T}}{}^{\overline{o}}\boldsymbol{R}^{p_{\overline{g}}\,\mathsf{T}}\boldsymbol{R}_{\overline{g}},
$$

$$
{}^{o}\boldsymbol{R}^{p_g} \doteq \boldsymbol{R}_{\overline{g}}{}^{o}\boldsymbol{R}^{p}{}^{p}\boldsymbol{R}^{p_{\overline{g}}} \quad | \quad {}^{p}\mathbf{I} \equiv {}^{p_g}\mathbf{I}_{g} \equiv {}^{p_{\overline{g}}}\overline{\mathbf{I}}_{\overline{g}}. \tag{12a}
$$

What eq. (12a) states is that in order for the reflected and transformed bodies to have equivalent angular kinetic energy, both bodies should have co-linear (or aligned) principal axes of inertia. This allows us to describe ${}^{o}\boldsymbol{R}^{p_g}$ as a function of the original body configuration ${}^{o}\boldsymbol{R}^{p}$ and two reflection matrices: the true reflection of space $\boldsymbol{R}_{\overline{g}}$ and a body specific diagonal reflection matrix ${}^{p}\boldsymbol{R}^{p_{\overline{g}}}$, which exists only if the rigid body has a symmetric mass distribution. A visual example for symmetric and asymmetrical rigid bodies is presented in supp.fig 4 left and middle columns.

### C.3 MODULARITY IN KINEMATIC TREES

As previously described, body $i$ of the real robot should have a reflected equivalent body $k$ with the same CoM position and aligned principle axes of inertia, up to a reflection of any of these axes. If body $i$ is unique in the kinematic tree, then $k = i$ and the body must have at least a symmetric mass distribution. When body $i$ is not unique (i.e. there exists another body $k \neq i$ that is its true reflected version), then no constraint of symmetric mass distribution is imposed on $i$, only the alignment of the principal axes of inertia is required. A didactic example of this scenario is presented in supp.fig 4-right, for Bolt's legs in supp.fig 5, and for Solo's legs in fig. 1

### C.4 SYMMETRIC POSITION AND VELOCITY CONSTRAINT CONFIGURATION SPACES

Although it is implicitly implied on eq. (2) that the constrained position Q and velocity $T_qQ$ configuration vector spaces should also be symmetric or equivariant, this property might be easily overlooked. As mentioned in section 4 the relevance of morphological symmetries relies on the equivariant nature of the system dynamics (eqs. (1) and (3)), which imprints symmetry constraints on optimal control policies and proprioceptive and exteroceptive measurements. However, with non-symmetric constrained configuration spaces, eq. (2) will not hold for every system state $q \in Q$, $\dot{q} \in T_qQ$, and any uncontrolled or controlled trajectory of the system dynamics shall not have a symmetric equivalent trajectory, as this has the potential to violate the constraints of the configuration space.

## D EFFICIENT CONSTRUCTION OF $\mathcal{G}$-EQUIVARIANT NNS FOR DISCRETE MORPHOLOGICAL SYMMETRY (DMS) GROUPS $\mathcal{G}$

As mentioned in section 5 our work builds upon the framework for the construction of $\mathcal{G}$-equivariant NN of Finzi et al. (2021b). The core limitation of this framework is the inability to handle large dimensional spaces, due to the computational and memory complexities. For instance, for an equivariant layer with input dimension $n$ and output dimension $m$, the computational complexity of finding the equivariant linear map basis **B** (which is quadratic $\mathcal{O}((mn)^2r^2)$ through the Krylov subspace method) and the memory complexity of **B** $\in \mathbb{R}^{mn \times r} \mid r \leq mn$, become easily intractable for moderate $n$ and $m$ dimensions. This limitation is openly discussed in the EMLP repository *README.md*, but regretfully not in the original paper.

In practice, we found these limitations when trying to construct the equivariant version of the Contact CNN (Lin et al., 2021) in our second experiment. This architecture in its internal layers has $n, m > 2000$, for which: (i) the Krylov subspace method complexity renders the operation intractable with standard hardware and (ii) the matrices **B** of internal layers required storage of $1[Gb] >$ for moderate input output dimensions ($m, n \approx 250$) and $1[Pb] >$ for $m, n > 2000$). See supp.table 1 for a comparison between dense and sparse matrix representations.

### D.1 TRAINABLE PARAMETER REDUCTION OF $\mathcal{G}$-EQUIVARIANT LAYERS (FOR $\mathcal{G}$ A DMS GROUP)

Determining analytically the number of trainable parameters (i.e. the rank $r$) of an $\mathcal{G}$-equivariant layer is, in general, an unresolved problem. However, for DMS groups, $r$ can be computed once the input-output action representations are known. The requirement to compute $r$ is that actions affecting the linear maps are a semi-direct product[10] of the input-output groups, and the input-output representations are generalized permutation matrices. These conditions are met for most DMS groups (see supplementary B).

The equivariance constraints of eq. (9) on linear maps of perceptron (or convolutional) layers imply a reduction of trainable parameters from $|\boldsymbol{w}| = mn$ to $|\boldsymbol{c}| = r \leq mn$. For DMS groups, $r$ is associated with the number of unique orbits of the elements of $\boldsymbol{w}$. Thus we can compute this value using the orbit-counting theorem (also known as Burnside's Lemma), which states that the number of orbits is the average number of fix-points of $\mathcal{G}$, that is $r = \frac{1}{|\mathcal{G}|} \sum_{g \in \mathcal{G}} |\boldsymbol{w}^g|$, where $\boldsymbol{w}^g \doteq \{w \in \boldsymbol{w} : g \cdot w = w\}$ represents the set of elements of $\boldsymbol{w}$ that are invariant to $g$ (i.e. fix-points). Those fix-points can be identified by the elements on the diagonal of $\rho_{\boldsymbol{w}}(g)$ that are equal to one. Therefore,

**Tri-Finger Robot $\mathcal{G} = \mathcal{C}_3$**

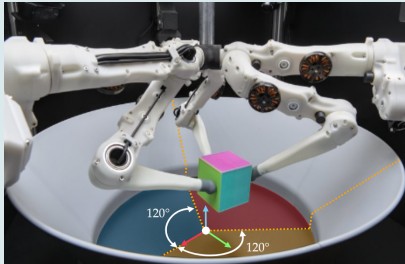

This fixed-based robot is symmetric w.r.t. rotations of space by $\theta = 120°$ in the vertical axis. Therefore, its symmetry group is the cyclic group of order three ($\mathcal{G} = \mathcal{C}_3$). To identify this symmetry group we apply the procedure in section 4.3:

1. **Identify $\boldsymbol{X}_B$ and $\mathbf{I}_B$**: As a fix-based robot, we define $\boldsymbol{X}_B$ to be the mounting structure supporting each finger, and the gray disk delimiting the workspace (see image).
2. **Identify symmetries of $\mathbf{I}_B$**: The inertia of this virtual base $\mathbf{I}_B$ is invariant to rotations by $120°$ in the vertical axis. I.e., $\mathbf{I}_B$ is invariant to $\boldsymbol{X}_B \rho_{\mathbb{E}_3}(\overline{g})^{-1} | \overline{g} \in \{e, \overline{g}_\theta, \overline{g}_\theta^2\} \equiv \mathcal{C}_3$ (eq. (6)).
3. **Identify modularity in the kinematic tree**: There are three symmetric kinematic subchains. Each finger is composed of replicated versions of the same bodies.
4. **Identify the DMS group $\mathcal{G}$**:

Consider that the transformation $\rho_{\mathbb{E}_3}(g)\boldsymbol{X}_B \doteq \rho_{\mathbb{E}_3}(\overline{g})\boldsymbol{X}_B\rho_{\mathbb{E}_3}(\overline{g})^{-1}$ (eq. (6)) can be interpreted as a rotation of the virtual base by $\theta°$ followed by a rotation $-\theta°$ in the $z$ axis. Thus respecting the fix-base constraint of the system. Denote the joint-space $\boldsymbol{q} = \hat{\boldsymbol{q}} = [\boldsymbol{q}_{f1}^{\mathsf{T}}, \boldsymbol{q}_{f2}^{\mathsf{T}}, \boldsymbol{q}_{f3}^{\mathsf{T}}]^{\mathsf{T}}$ be composed of each finger's DoF ($\boldsymbol{q}_{fi} \in \mathbb{R}^3$).

Then we can define $\rho_{\mathbb{Q}_J}(g) \doteq \rho_{\mathbb{R}^3}(\overline{g}) \otimes \boldsymbol{I}_3 | \overline{g} \in \mathcal{C}_3$. Being $\rho_{\mathbb{R}^3}(\cdot)$ the unique representation of the actions of $\mathcal{C}_3$ on a 3-dimensional space (3 subchains). For the generator action of the group this is $\rho_{\mathbb{R}^3}(\overline{g}) = \begin{bmatrix} 0 & 1 & 0 \\ 0 & 0 & 1 \\ 1 & 0 & 0 \end{bmatrix}$.

Lastly, we verify if $\mathcal{G} = \mathcal{C}_3$ by testing all tentative group actions for DMSs eq. (5).

**Augmentation of data samples**: Say we collect a dataset of robot states $(\boldsymbol{q}, \dot{\boldsymbol{q}})$ and cube states $\boldsymbol{X}_C$ at every time step $t$, to train the manipulation policy (Funk et al., 2021). To obtain the symmetric states, at every $t$, we need to understand that since we are imitating the effect of a true rotation of space $\overline{g}$, the symmetric states are obtained by $(g \cdot \boldsymbol{q}, g \cdot \dot{\boldsymbol{q}})$ and $(\overline{g} \cdot \boldsymbol{X}_C \doteq \rho_{\mathbb{E}_3}(\overline{g})\boldsymbol{X}_C)$.

**Bolt Bipedal Robot $\mathcal{G} = \mathcal{C}_2$**

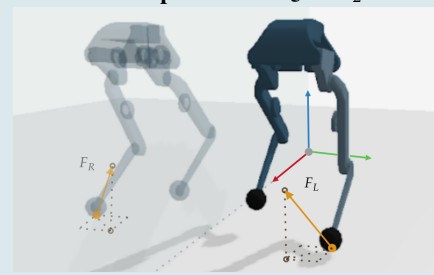

Bolt is a bipedal robot with a sagittal plane reflection symmetry ($\mathcal{G} = \mathcal{C}_2$). This morphological symmetry allows it to imitate the effect of arbitrary reflections of space ($\overline{g} \in \mathbb{E}_3$) by re-configuring its base and legs. To identify this symmetry group we apply the procedure in section 4.3:

1. **Identify $\boldsymbol{X}_B$ and $\mathbf{I}_B$**: $\boldsymbol{X}_B$ is the robot base (hips) body, with its corresponding inertia $\mathbf{I}_B$
2. **Identify symmetries of $\mathbf{I}_B$**: The base body has symmetrical mass distribution w.r.t the sagittal plane. Thus, $\mathbf{I}_B$ is invariant to the transformation $\boldsymbol{X}_B \rho_{\mathbb{E}_3}(\overline{g})^{-1} | \overline{g} \in \{e, \overline{g}_s\} \equiv \mathcal{C}_2$ (eq. (6)).
3. **Identify modularity in the kinematic tree**: There are two symmetric kinematic subchains. The left leg subchain and bodies are reflected versions of the right leg subchain and bodies.
4. **Identify the DMS group $\mathcal{G}$**:

Since a reflection w.r.t to the sagittal plane would imply a true reflection of the rigid bodies of the legs, we need to *permute* each body in the kinematic tree with each reflected version. Denote the joint-space $\hat{\boldsymbol{q}} = [\boldsymbol{q}_L^{\mathsf{T}}, \boldsymbol{q}_R^{\mathsf{T}}]^{\mathsf{T}}$ as composed of the left $L$ and right $R$ legs' DoF ($\boldsymbol{q}_{L/R} \in \mathbb{R}^3$). Denote the sign-relation between the DoF of the Left and right legs' degrees of freedom as $\boldsymbol{s}_{L|R} \in \mathbb{R}^3$.

Then we can define $\rho_{\mathbb{Q}_J}(g) \doteq \rho_{\mathbb{R}^2}(\overline{g}) \otimes (\boldsymbol{s}_{L|R}\boldsymbol{I}_3) | \overline{g} \in \mathcal{C}_2$. Being $\rho_{\mathbb{R}^2}(\cdot)$ the unique representation of the actions of $\mathcal{C}_2$ on a 2-dimensional space (2 subchains). For the non-trivial action of the group this is $\rho_{\mathbb{R}^2}(\overline{g}_s) = \begin{bmatrix} 0 & 1 \\ 1 & 0 \end{bmatrix}$.

Lastly, recalling the definition of $\rho_{\mathbb{E}_3}(g)$ in eq. (6), we verify if $\mathcal{G} = \mathcal{C}_2$ by testing all tentative group actions for DMSs eq. (5).

**Augmentation of data samples**: Say we collect a dataset of robot states $(\boldsymbol{q}, \dot{\boldsymbol{q}})$ and ground reaction forces $(\boldsymbol{f}_L, \boldsymbol{f}_R)$, that we transform to the space of generalized forces as $(\boldsymbol{\tau}_{f_L}, \boldsymbol{\tau}_{f_R})$, at every timestep $t$. Aiming to train a reactive locomotion policy (Ordonez-Apraez et al., 2022). The symmetric states, at every $t$, are then defined as: $(g \cdot \boldsymbol{q}, g \cdot \dot{\boldsymbol{q}})$ and $(g \cdot \boldsymbol{\tau}_{f_L}, g \cdot \boldsymbol{\tau}_{f_R}) \equiv (\rho_{\mathbb{Q}}(g)\boldsymbol{\tau}_{f_L}, \rho_{\mathbb{Q}}(g)\boldsymbol{\tau}_{f_R})$

Supplementary Figure 5: **Example morphological symmetries of the Tri-Finger (Funk et al., 2021) and Bolt robots**, allowing to imitate a rotation of space (left) and a reflection of space (right).

Supplementary Table 1: **Comparison of memory complexity of individual layers of the equivariant version of Contact-CNN Lin et al. (2021) (ECNN).** This example compares the sparse and dense representations of matrices $\mathbf{B} \in \mathbb{R}^{mn \times r}$ and the $|\mathcal{G}|$ group action representations $\rho_w(g) \in \mathbb{R}^{mn \times mn}$, for the symmetry group $\mathcal{G} = \mathcal{C}_2$ of the Mini-Cheetah robot, with $r = {mn}/{2}$ (see supp.eq 13). Here, $n, m$ represents the input and output dimensions of each layer. The dense memory complexity of all action representations increases with the group order $|\mathcal{G}|$ while the memory complexity for $\mathbf{B}$ decreases with larger group orders (since $r \le mn$ becomes smaller). We assume floating point representations with 32 bits.

| | | | Dense Memory [Bytes] | | Sparse Memory [Bytes] | |
|---|---|---|---|---|---|---|
| **Layer Type** | $n$ | $m$ | $\rho_w(g)$ | $\mathbf{B}$ | $\rho_w(g)$ | $\mathbf{B}$ |
| 1D-Conv | 54 | 64 | $764.41M$ | $191.10M$ | $221.18k$ | $110.59k$ |
| 1D-Conv | 64 | 64 | $1.07G$ | $268.43M$ | $262.14k$ | $131.07k$ |
| 1D-Conv | 64 | 128 | $4.29G$ | $1.07G$ | $524.28k$ | $262.14k$ |
| 1D-Conv | 128 | 128 | $17.18G$ | $4.29G$ | $1.04M$ | $524.28k$ |
| Percept | 4736 | 2048 | $6.02P$ | $1.50P$ | $620.75M$ | $310.37M$ |
| Percept | 2048 | 512 | $70.36T$ | $17.59T$ | $67.10M$ | $33.55M$ |
| Percept | 512 | 16 | $4.29G$ | $1.07G$ | $524.28k$ | $262.14k$ |

for a $\mathcal{G}$-equivariant layer, the number of trainable parameters is determined by:

$$r = \tfrac{1}{|\mathcal{G}|}\textstyle\sum_{g \in \mathcal{G}} \chi^{\mathbf{1}}_{\rho_{\boldsymbol{W}}}(g) = \tfrac{1}{|\mathcal{G}|}\textstyle\sum_{g \in \mathcal{G}} \chi^{\mathbf{1}}_{\rho_{in}}(g^{-1}) \cdot \chi^{\mathbf{1}}_{\rho_{out}}(g), \tag{13}$$

denoting $\chi^{\mathbf{1}}_{\rho}(g) : \mathcal{G} \to \mathbb{N}$ as the number of fix-points of the action representation $\rho(g)$. Therefore, the number of trainable parameters can range from ${|\boldsymbol{w}|}/{|\mathcal{G}|} \le r \le |\boldsymbol{w}|$, depending on the fix-points of the layers' input and output spaces.

### D.2 PARAMETER INITIALIZATION OF EQUIVARIANT LAYERS FOR DMS

Consider a Equivariant Neural Network architecture composed of multiple layers of equivariant linear (or convolutional) layers of the form $^l\boldsymbol{y} := \sigma(^l\boldsymbol{W}\,^l\boldsymbol{x} + \,^l\boldsymbol{b})$, being $l$ the layer index, $^l\boldsymbol{x} \in \mathbb{R}^n$ and $^l\boldsymbol{y} \in \mathbb{R}^m$ the layer's input and output vector spaces, $^l\boldsymbol{W} \doteq \sum_k^r \,^lc_k\,^l\mathbf{B}_{:,:,k} \in \mathbb{R}^{m \times n}$ the layer's linear map, $^l\mathbf{B} \in \mathbb{R}^{m \times n \times r}$ the layer's $r$ basis vectors spawning the space of equivariant linear maps, $^l\boldsymbol{c} \in \mathbb{R}^r$ the layer's trainable parameters, and $^l\boldsymbol{b} \in \mathbb{R}^m$ the layer's bias vector.

For the optimal flow of information throughout the network, it's relevant to initialize the trainable parameters such that the variance of activations (during inference/forward-propagation) and gradients (during back-propagation) is kept constant, avoiding activations/gradients from vanishing or exploiting (Glorot & Bengio, 2010)[11].

The derivation is based on the equivalent process for unconstrained layers presented in He et al. (2015). Let the layer's activations before the non-linearity be denoted by $^l\boldsymbol{z} = \,^l\boldsymbol{W}\,^l\boldsymbol{x} + \,^l\boldsymbol{b}$, such that $^l\boldsymbol{y} = \sigma(^l\boldsymbol{z})$, and note that $^l\boldsymbol{x} = \,^{l-1}\boldsymbol{y}$. Furthermore, we will assume the elements of $^l\boldsymbol{c}$ and $^l\boldsymbol{x}$ are mutually independent and sampled from two independent distributions, denoting the random variables of the two distributions as $^lc$ and $^l\mathrm{x}$.

---

[11]See Pierre Ouannes blog: https://pouannes.github.io/blog/initialization/#xavier-and-kaiming-initialization

The core difference in the initialization of unconstrained and equivariant layers lies in the way the linear map is parameterized. For equivariant layers we have:

$$\text{Var}(^l\boldsymbol{W}^l\boldsymbol{x} + {}^l\boldsymbol{b}) = \sum_i^m \sum_j^n \text{Var}\left(^lW_{i,j}\,{}^l\text{x}_j\right) \qquad |\ \text{Var}(^l\boldsymbol{b}) = 0$$

$$= \sum_i^m \sum_j^n \text{Var}\left(\left(\sum_k^r {}^lc_k\,{}^l\mathbf{B}_{m,n,k}\right){}^l\text{x}_j\right)$$

$$, = \text{Var}\left(^lc\,{}^l\text{x}\right) \underbrace{\sum_i^m \sum_j^n \sum_k^r \mathbf{B}_{m,n,k}^2}_{\lambda_{l\mathbf{B}}} \quad |\ \text{Var}\left(\sum_a \underset{const}{s_a}\ \mathbf{p}\right) = \sum_a s_a^2 \text{Var}(\mathbf{p})$$

$$(14)$$

In the forward-propagation scenario, we are interested in conserving the variance of the activations throughout layers, that is we must ensure $\text{Var}(^l\mathbf{z}) = \text{Var}(^{l-1}\mathbf{z})$. Using supp.eq 14 we obtain:

$$\text{Var}(^l\boldsymbol{z}) = \text{Var}(^l\boldsymbol{W}^l\boldsymbol{x} + {}^l\boldsymbol{b})$$

$$m\,\text{Var}(^l\boldsymbol{z}) = \lambda_{l\mathbf{B}}\text{Var}\left(^lc\,{}^l\text{x}\right)$$

$$\text{Var}(^l\mathbf{z}) = \frac{\lambda_{l\mathbf{B}}}{m}\left(\underbrace{\mathbb{E}(^lc^2)\mathbb{E}(^l\text{x}^2)}_{\text{Var}(^lc)} - \underbrace{\mathbb{E}(^lc)^2\mathbb{E}(^l\text{x})^2}_{=0}\right)$$

$$\text{Var}(^l\mathbf{z}) = \frac{\lambda_{l\mathbf{B}}}{m}\text{Var}(^lc)\mathbb{E}\left(^{l-1}\text{y}^2\right) \qquad\qquad |\ ^l\text{x} = {}^{l-1}\text{y} = \sigma(^{l-1}\mathbf{z})$$

$$\text{Var}(^l\mathbf{z}) = \frac{\lambda_{l\mathbf{B}}\lambda_\sigma}{m}\text{Var}(^lc)\text{Var}(^{l-1}\mathbf{z}) \qquad |\ \mathbb{E}\left(^{l-1}\text{y}^2\right) = \lambda_\sigma\text{Var}\left(^{l-1}\mathbf{z}\right) \quad(15)$$

$$\text{Var}(^l\mathbf{z}) \equiv \text{Var}(^{l-1}\mathbf{z}) \qquad\qquad\qquad |\ \text{Var}\left(^lc\right) \doteq \frac{m}{\lambda_{l\mathbf{B}}\lambda_\sigma} \quad(16)$$

where $\lambda_\sigma$ in supp.eq 15 is a non-linearity dependent scalar computed analytically or empirically (see He et al. (2015)). In supp.eq 16 we conclude that if we sample the equivariant layer trainable parameters $^l\boldsymbol{c}$ from a distribution ensuring $\text{Var}(^lc) \doteq \frac{m}{\lambda_{l\mathbf{B}}\lambda_\sigma}$, the variance of the activations across equivariant layers remain constant. A similar procedure can be applied to the backward propagation case, concluding that in order to maintain a constant variance of the gradients across the network layers we should sample the trainable parameters ensuring $\text{Var}(^lc) \doteq \frac{n}{\lambda_{l\mathbf{B}}\lambda_\sigma}$. As remarked in He et al. (2015) both variance values for the forward and backward propagation cases lead to the proper flow of information in the network. On supp.fig 8, it can be appreciated that our method achieves equivalent results for equivariant architectures as He et al. (2015) does for standard linear and convolutional architectures.

# E  IMPLEMENTATION DETAILS & CODE

Additional to this section, we provide open-access code with the scripts for reproducing the experiments of this work, the parameters of the models used for comparison, along with additional interactive examples visualizing morphological symmetries of both robotic systems and data.

## E.1  EFFICIENT DATA AUGMENTATION

Since any input $\boldsymbol{x}$ and output $\boldsymbol{y}$ spaces of equivariant architectures have matrix symmetry action representations, $\rho_{\boldsymbol{x}}(g)$, $\rho_{\boldsymbol{y}}(g)$, it is possible to perform batched data augmentation, reducing the computational complexity of augmenting a batch of $N_b$ samples from $N_b$ matrix-vector multiplications to a single matrix-matrix multiplication, preferably performed after data is loaded to GPU for optimal performance.

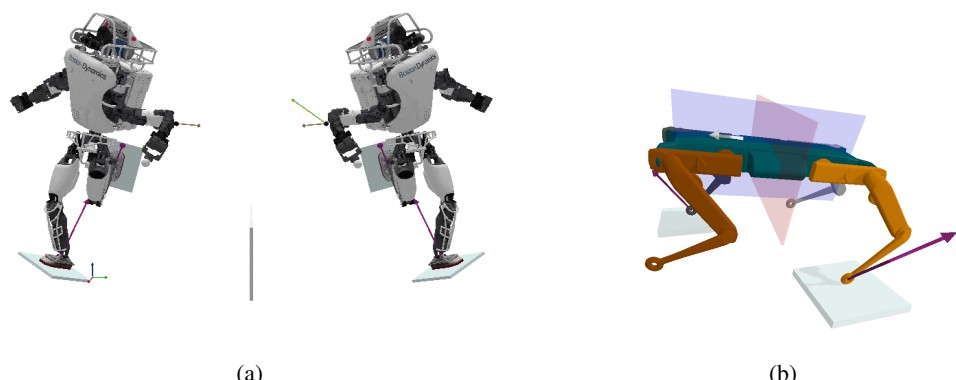

(a)                        (b)

Supplementary Figure 6: (a) Atlas robot $\mathcal{C}_2$ symmetry affecting proprioceptive ($q$, $\dot{q}$, $\ddot{q}$, $\tau$) and exteroceptive measurements (ground reaction forces, terrain surface and height estimations) (see 3D-animation). (b) Robot Solo sagittal (blue) and transversal (red) symmetry planes of the base rigid body, allowing the system to possess $\mathcal{K}_4$ symmetry

## E.2    HYPERPARAMETER TUNNING

The only hyper-parameter tunned for each model and model variant was the learning rate. For all model variants presented in this work (except the original Contact-CNN model from Lin et al. (2021), which we retrained using the same hyperparameters reported by the authors) we ran a grid-search in log-scale among 20 different learning rates. In this scenario, we always used the entire training dataset and optimized w.r.t computed loss in the entire validation partition. The learning rate values used for each model are depicted in supp.table 2.

## E.3    EXPERIMENT: COM MOMENTUM ESTIMATION

The dataset for the CoM estimation experiment was generated using Pinocchio (Carpentier et al., 2019), which in turn uses the URDF models of the robots Solo and Atlas, to extract the kinematic and dynamic parameters required to compute the Centroidal Momentum Matrix $\mathbf{A}_G(q)$ matrix (Orin et al., 2013), with which computing the CoM momentum reduces to:

$$g \cdot h = \mathbf{A}_G(g \cdot \hat{q})g \cdot \dot{\hat{q}} \qquad | \quad \forall g \in \mathcal{G}. \qquad (17)$$

$$g \cdot h \approx f\left(g \cdot \hat{q}, g \cdot \dot{\hat{q}}; \phi\right) \qquad | \quad \forall g \in \mathcal{G}. \qquad (18)$$

Where supp.eq 17 expresses the analytical $\mathcal{G}$-equivariant function to compute the CoM momentum. While supp.eq 18 is the approximation of this function by an $\mathcal{G}$-equivariant NN, with parameters $\phi$.

### E.3.1    DETERMINATION OF THE INPUT AND OUTPUT REPRESENTATIONS $\rho_{\boldsymbol{x}}(g), \rho_{\boldsymbol{y}}(g) \mid g \in \mathcal{G}$

Both robots Solo and Atlas evolve in the Euclidean space of 3-dimensions. Therefore their configuration space can be decoupled into $Q \doteq \mathbb{E}_3 \times Q_J$. After identifying their symmetry groups and their corresponding $\mathbb{E}_3$ and $Q_J$ representations ($\rho_{\mathbb{E}_3}(g), \rho_Q(g) \mid \forall\, g \in \mathcal{G}$), identifying the representations of the input and output spaces of the NN function approximator (supp.eq 18) becomes a trivial task considering that:

$$\underbrace{\begin{bmatrix} \rho_{\mathbb{E}_d}(\overline{g}) & \mathbf{0} \\ \mathbf{0} & \rho_{\mathbb{E}_d}(\overline{g}) \end{bmatrix}}_{\rho_{\boldsymbol{y}}(g)} \underbrace{\begin{bmatrix} l \\ \mathbf{k} \end{bmatrix}}_{\boldsymbol{y}} \approx f\left( \underbrace{\begin{bmatrix} \rho_{Q_J}(g) & \mathbf{0} \\ \mathbf{0} & \rho_{Q_J}(g) \end{bmatrix}}_{\rho_{\boldsymbol{x}}(g)} \underbrace{\begin{bmatrix} \hat{q} \\ \dot{\hat{q}} \end{bmatrix}}_{\boldsymbol{x}} \right) \qquad | \quad \forall \quad (g, \overline{g})|g \in \mathcal{G}, \overline{g} \in \mathbb{E}_3 \quad (19)$$

Defining $\boldsymbol{x} = \begin{bmatrix} \hat{q} \\ \dot{\hat{q}} \end{bmatrix} \in \mathbb{R}^{2n_J}$ and $\boldsymbol{y} = \boldsymbol{h} \in \mathbb{R}^{2d} \equiv \mathbb{R}^6$. Note that by definition any improper transformation applied to a pseudo-vector (e.g. angular velocity/momentum, torques) is computed as $|\boldsymbol{R}|\boldsymbol{R} \cdot \mathbf{k}$.

### E.3.2 PRACTICAL DETAILS OF THE DATASET GENERATION

The URDF files of the robots Solo and Atlas are generated using XACRO scripts, which replicate the structure of limbs to their symmetric counterparts, making the dynamics of the robots in simulation exactly $\mathcal{G}$-equivariant. However, the algorithm for computing the CoM momentum from Pinocchio is numerically sensitive, resulting in the orbits of the momentum $\mathcal{G} \cdot \boldsymbol{h}$ deviating slightly from the theoretical orbits. Therefore to reduce numerical errors and ensure the theoretical equivariance of the data, we replace every target variable by the average of its orbit $\boldsymbol{y} = \boldsymbol{h} \doteq \frac{1}{|G|} \sum \mathcal{G} \cdot g^{-1}(\mathbf{A}_G \left( \rho_Q(g)\hat{\boldsymbol{q}} \right) \rho_Q(g)\dot{\hat{\boldsymbol{q}}}) \quad | \quad \forall \quad g \in \mathcal{G}$.

### E.4 EXPERIMENT: STATIC-FRICTION-REGIME CONTACT DETECTION

The dataset presented in (Lin et al., 2021) is composed of output samples $\boldsymbol{y} \in \mathbb{R}^{16}$, where each dimension of $\boldsymbol{y}$ represents a logit of a specific contact state, among the 16 different combinations of each of the 4 legs possible binary contact states. The input samples $\boldsymbol{y} = \{\boldsymbol{z}_i\}_{i=0}^{150} \in \mathbb{R}^{54 \times 150}$, are a history of 150 samples $\boldsymbol{z} = [\hat{\boldsymbol{q}}, \dot{\hat{\boldsymbol{q}}}, \boldsymbol{a}, \boldsymbol{w}, \boldsymbol{p}, \boldsymbol{v}] \in \mathbb{R}^{54}$. Where $\hat{\boldsymbol{q}} \in \mathbb{R}^{n_J}, \dot{\hat{\boldsymbol{q}}} \in \mathbb{R}^{n_J}, \boldsymbol{a}\mathbb{R}^3, \boldsymbol{w} \in \mathbb{R}^3, \boldsymbol{p} \in \mathbb{R}^{12}, \boldsymbol{v} \in \mathbb{R}^{12}$ are the MIT-Mini-Cheetah robot joint-space positions, velocities, base linear acceleration, base angular velocity, and each of the four legs feet's position and velocities, respectively, referenced to the robots base frame $B$.

The function approximator to learn is expected to be approximately equivariant to the reflection group $\mathcal{C}_2$, considering the sagittal symmetry of the robot morphology. Therefore:

$$g \cdot \boldsymbol{y} = f\left( g \cdot \boldsymbol{x}; \phi \right) \quad | \quad g \in \mathcal{G} = \mathcal{C}_2 \tag{20}$$

### E.4.1 DETERMINATION OF THE INPUT AND OUTPUT REPRESENTATIONS $\rho_{\boldsymbol{x}}(g), \rho_{\boldsymbol{y}}(g) \mid g \in \mathcal{G}$

The MiniCheetah robot evolves in the Euclidean space of 3-dimensions. Therefore its configuration space can be decoupled into $Q \doteq \mathbb{E}_3 \times Q_J$. After identifying their symmetry groups and their corresponding $\mathbb{E}_3$ and $Q_J$ representations $(\rho_{\mathbb{E}_3}(g), \rho_Q(g) \mid \forall g \in \mathcal{G})$, we can identify the representations of the input and output spaces of the NN function approximator (supp.eq 20), considering that:

$$\rho_{\boldsymbol{y}}(g)\boldsymbol{y} \approx f\left( \underbrace{\begin{bmatrix} \rho_{Q_J}(g) & \mathbf{0} & \mathbf{0} & \mathbf{0} & \mathbf{0} & \mathbf{0} \\ \mathbf{0} & \rho_{Q_J}(g) & \mathbf{0} & \mathbf{0} & \mathbf{0} & \mathbf{0} \\ \mathbf{0} & \mathbf{0} & \rho_{\mathbb{E}_3}(\overline{g}) & \mathbf{0} & \mathbf{0} & \mathbf{0} \\ \mathbf{0} & \mathbf{0} & \mathbf{0} & \rho_{\mathbb{E}_3}(\overline{g}) & \mathbf{0} & \mathbf{0} \\ \mathbf{0} & \mathbf{0} & \mathbf{0} & \mathbf{0} & \rho_{\boldsymbol{p}}(g) & \mathbf{0} \\ \mathbf{0} & \mathbf{0} & \mathbf{0} & \mathbf{0} & \mathbf{0} & \rho_{\boldsymbol{p}}(g) \end{bmatrix}}_{\rho_{\boldsymbol{x}}(g)} \underbrace{\begin{bmatrix} \hat{\boldsymbol{q}} \\ \dot{\hat{\boldsymbol{q}}} \\ \boldsymbol{a} \\ \boldsymbol{w} \\ \boldsymbol{p} \\ \boldsymbol{v} \end{bmatrix}}_{\boldsymbol{x}} \right) | \forall (g, \overline{g}) | g \in \mathcal{G}, \overline{g} \in \mathbb{E}_3$$

Where the representation $\rho_{\boldsymbol{p}}(g)$ acting on $\boldsymbol{p} \in \mathbb{R}^{12}$ and $\boldsymbol{v} \in \mathbb{R}^{12}$ is determined understanding that each of the feet positions $(\boldsymbol{p}_{RF}, \boldsymbol{p}_{LF}, \boldsymbol{p}_{RH}, \boldsymbol{p}_{LH})$ and velocities $(\boldsymbol{v}_{RF}, \boldsymbol{v}_{LF}, \boldsymbol{v}_{RH}, \boldsymbol{v}_{LH})$ are simply vectors living in $\mathbb{E}_3$. Thus, we must apply the euclidean action $\rho_{\mathbb{E}_3}(\overline{g})$ while at the same time permuting the feets (similar to the permutation of the kinematic subchains described by $\rho_{Q_J}(g)$):

$$g \cdot \boldsymbol{p} = \underbrace{\rho_{\mathbb{R}^4}(g) \otimes \rho_{\mathbb{E}_3}(\overline{g})}_{\rho_{\boldsymbol{p}}(g)}\boldsymbol{p}, \qquad g \cdot \boldsymbol{v} = \underbrace{\rho_{\mathbb{R}^4}(g) \otimes \rho_{\mathbb{E}_3}(\overline{g})}_{\rho_{\boldsymbol{p}}(g)}\boldsymbol{v} \qquad | \forall (g, \overline{g}) | g \in \mathcal{G}, \overline{g} \in \mathbb{E}_3 \tag{21}$$

$$= \begin{bmatrix} 0 & 1 & 0 & 0 \\ 1 & 0 & 0 & 0 \\ 0 & 0 & 0 & 1 \\ 0 & 0 & 1 & 0 \end{bmatrix} \otimes \rho_{\mathbb{E}_3}(\overline{g}) \begin{bmatrix} \boldsymbol{p}_{RF} \\ \boldsymbol{p}_{LF} \\ \boldsymbol{p}_{RH} \\ \boldsymbol{p}_{LH} \end{bmatrix}, \qquad = \begin{bmatrix} 0 & 1 & 0 & 0 \\ 1 & 0 & 0 & 0 \\ 0 & 0 & 0 & 1 \\ 0 & 0 & 1 & 0 \end{bmatrix} \otimes \rho_{\mathbb{E}_3}(\overline{g}) \begin{bmatrix} \boldsymbol{v}_{RF} \\ \boldsymbol{v}_{LF} \\ \boldsymbol{v}_{RH} \\ \boldsymbol{v}_{LH} \end{bmatrix} \quad | (g, \overline{g}) | \mathcal{C}_2 = \{e, g\} \tag{22}$$

Being $\rho_{\mathbb{R}^4}(g) \mid \forall g \in \mathcal{G}$ the unique representations of the finite group $\mathcal{G}$ in a 4-dimensional space, representing the 4 kinematic tree's subchains. This representation, for the non-trivial action of the MiniCheetah DMS group $\mathcal{G} \approx \mathcal{C}_2$ is expressed in supp.eq 22. The nature of this representation might be better understood if you consider that $\rho_{Q_J}(g) \doteq \rho_{\mathbb{R}^4}(g) \otimes \boldsymbol{I}_{n_s}$, being $n_s$ the number of degrees of freedom of the kinematic subchain (which for the case of MiniCheetah is 3 DoF). See simpler examples in supp.fig 5.

Lastly, the representation for the contact state $\rho_{\boldsymbol{y}}(g)$ is given by the permutation matrix relating $\boldsymbol{y}$ and $g \cdot \boldsymbol{y}$ described in supp.table 3.

Supplementary Table 2: Robot and Model parameters for CoM and Contact state estimation.

| Robot | Model | $\mathcal{G}$ | lr | Data Samples |
|---|---|---|---|---|
| Solo | MLP/EMLP | $\mathcal{C}_2/\mathcal{K}_4$ | $2.4 \times 10^{-3}$ | 100k |
| Atlas | MLP/EMLP | $\mathcal{C}_2$ | $1.5 \times 10^{-3}$ | 500k |
| MIT-MiniCheetah | CNN | $\mathcal{C}_2$ | $1.0 \times 10^{-4}$ | 730k |
| MIT-MiniCheetah | E-CNN | $\mathcal{C}_2$ | $1.0 \times 10^{-5}$ | |

### E.4.2 DETAILS ON DATASET PARTITIONING

We modified the original dataset partitioning to properly evaluate the generalization capacity of the models. The original dataset was composed of 15 different recordings varying ground type and gait type used during data collection (most recordings were performed on a trot gait, which heavily biased the dataset to contact states 0, 6, and 9 of supp.table 3. See supp.fig 7).

The authors of (Lin et al., 2021) partitioned all 15 recordings into $(70\%, 15\%, 15\%)$ training, validation and testing. This partition was made such that the first $70\%$ time-samples of each recording were assigned for training, the following $15\%$ to validation, and the rest for testing.

Because we are interested in studying the generalization capacity of the models and the out-of-training-distribution performance, we modified this partitioning such that among the 15 different recordings we selected randomly 5 recordings for testing, and the remaining 10 recordings were used for training splitting these recordings into $(85\%, 15\%)$ training and validation splits as in Lin et al. (2021), that is, for each recording, the first $85\%$ data-samples go for training and the remaining for validation.

The selected training-validation recordings were: air_walking_gait, concrete_difficult_slippery, concrete_left_circle, middle_pebble, rock_road, asphalt_road, concrete_galloping, grass, old_asphalt_road, sidewalk. While the selected testing recordings were: air_jumping_gait, concrete_pronking, concrete_right_circle, forest, small_pebble.

### E.5 MITIGATION OF SUBOPTIMAL ASYMMETRIES IN MODEL PERFORMANCE

When comparing individual leg classification we see that the equivariant model converges to having a similar performance for each symmetric pair of legs, while the unconstrained models converge to an asymmetrical suboptimal state favoring the contact detection of one leg at the expense of reduced performance for the symmetric leg (see LF and RF f1-scores). This asymmetrical performance is attributed to the CNN and CNN-aug models learning to extract temporal features for both symmetric legs separately, increasing the likelihood of converging to asymmetrical local minima. On the contrary, the equivariant model E-CNN can be thought of as learning to extract a single set of *symmetric* temporal features for each symmetric pair of states (a consequence of the model's equivariance and parameter sharing). This implies that the temporal features used for determining the contact state of, say the left frontal leg, would also be used to determine the contact state of the symmetric leg, the right frontal leg, and vice-versa.

### E.6 EQUIVARIANT CONV1D LAYERS

For details on the construction of the Equivariant 1D Convolutional layers reefer to [2]. Note that the symmetry of a single time-sample $z_i$ is shared across all time-samples $y = \{z_i\}_{i=0}^{150}$.

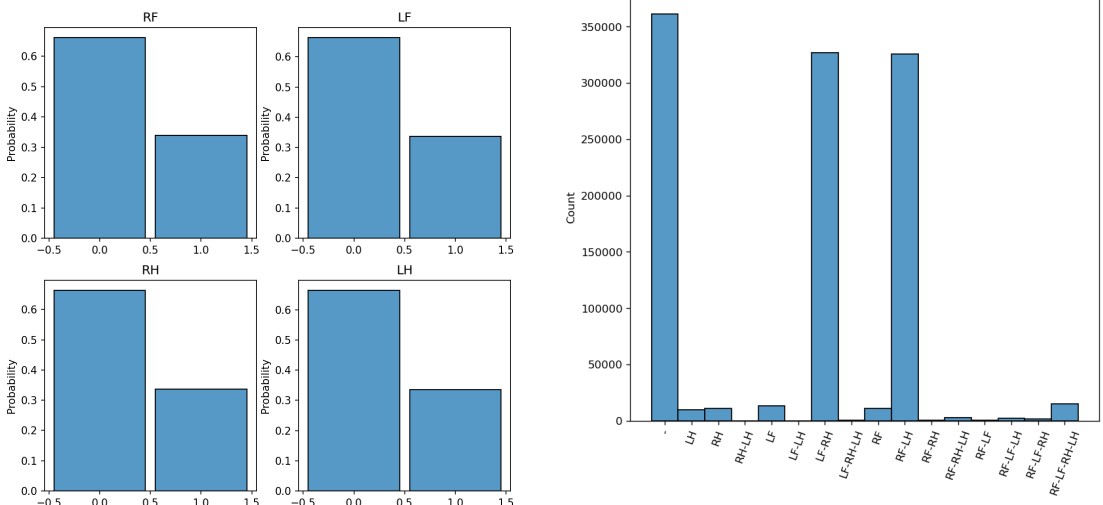

Supplementary Figure 7: Sample distribution of the training set of the contact dataset presented by Lin et al. (2021). The left figure shows the distribution of the binary contact state for each of the mini-cheetah legs, clearly displaying a similar distribution among legs favoring states of no contact (approximately for each leg $60\%$ of data samples represent states of no contact). The right figure shows the distribution of contact states ($\boldsymbol{y} \in \mathbb{R}^{16}$), clearly showing the dataset imbalance caused by the majority of the data being collected by the robot performing a trot gait.

Supplementary Table 3: Symmetric contact state for Mini-Cheetah quadruped robot, considering its sagittal symmetry, and morphological symmetry group $\mathcal{C}_2$. Each leg binary contact state (LF: Left Front, RF: Right Front, LH: Left Hind, RH: Right Hind) is displayed with its corresponding robot contact state $\boldsymbol{y}$.

| RF | LF | RH | LH | $\boldsymbol{y}$ | $g \cdot \boldsymbol{y}$ | LF | RF | LH | RH |
|----|----|----|----|---|-----|----|----|----|----|
| 0 | 0 | 0 | 0 | 0 | 0 | 0 | 0 | 0 | 0 |
| 0 | 0 | 0 | 1 | 1 | 2 | 0 | 0 | 1 | 0 |
| 0 | 0 | 1 | 0 | 2 | 1 | 0 | 0 | 0 | 1 |
| 0 | 0 | 1 | 1 | 3 | 3 | 0 | 0 | 1 | 1 |
| 0 | 1 | 0 | 0 | 4 | 8 | 1 | 0 | 0 | 0 |
| 0 | 1 | 0 | 1 | 5 | 10 | 1 | 0 | 1 | 0 |
| 0 | 1 | 1 | 0 | 6 | 9 | 1 | 0 | 0 | 1 |
| 0 | 1 | 1 | 1 | 7 | 11 | 1 | 0 | 1 | 1 |
| 1 | 0 | 0 | 0 | 8 | 4 | 0 | 1 | 0 | 0 |
| 1 | 0 | 0 | 1 | 9 | 6 | 0 | 1 | 1 | 0 |
| 1 | 0 | 1 | 0 | 10 | 5 | 0 | 1 | 0 | 1 |
| 1 | 0 | 1 | 1 | 11 | 7 | 0 | 1 | 1 | 1 |
| 1 | 1 | 0 | 0 | 12 | 12 | 1 | 1 | 0 | 0 |
| 1 | 1 | 0 | 1 | 13 | 14 | 1 | 1 | 1 | 0 |
| 1 | 1 | 1 | 0 | 14 | 13 | 1 | 1 | 0 | 1 |
| 1 | 1 | 1 | 1 | 15 | 15 | 1 | 1 | 1 | 1 |

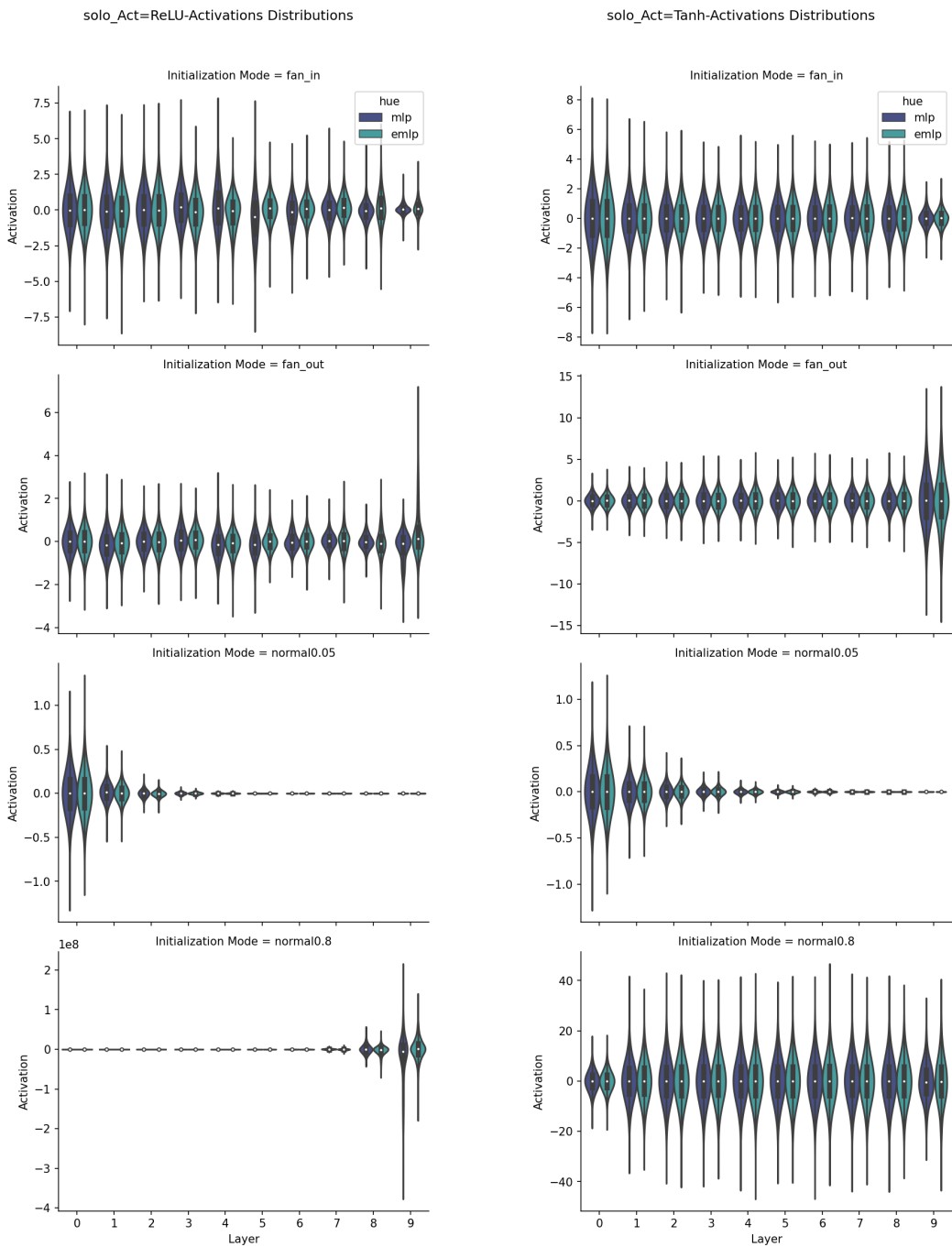

Supplementary Figure 8: Comparison of the initialization method of unconstrained layers of He et al. (2015) with our initialization method for equivariant layers. Left and right columns correspond to MLP and EMLP architectures with $\sigma = ReLu$ and $\sigma = Tanh$ non-linearities, respectively. Each row shows different initialization methods differing in the variance of the initialization distribution of the layer's trainable parameters. First and second rows show the forward and backward propagation cases of He et al. (2015) for MLP and of supplementary D.2 for EMLP, with $\mathrm{Var}(^l c) \doteq {}^m/_{(\lambda_{l_{\mathbf{B}}} \lambda_\sigma)}$ and $\mathrm{Var}(^l c) \doteq {}^n/_{(\lambda_{l_{\mathbf{B}}} \lambda_\sigma)}$, respectively. In these cases, the variance of activations through the network depth remains nearly constant, as desired. The last two rows show the initialization of layer parameters with a constant variance of $0.05^2$ and $0.8^2$, illustrating scenarios of activations vanishings and exploiting. All architectures are composed of 10-layers with 256 neurons on intermediate layers. In the equivariant case, the network is $\mathcal{K}_4$-equivariant.

