# OpenReview forum: "On discrete symmetries of robotics systems: A group-theoretic and data-driven analysis"
_ICLR.cc/2023/Conference — Submitted to ICLR 2023_

### Official Review · Reviewer_tbqU · 2022-10-19

**Confidence:** 3
**Correctness:** 3
**Technical Novelty And Significance:** 2
**Empirical Novelty And Significance:** 3
**Recommendation:** 5

**Clarity, Quality, Novelty And Reproducibility:**

Quality: The definition and expression can be improved.

Novelty: The problem statement and the proposed method seem straightforward combinations of existing studies. It is better to what is non-trivial.


**Strength And Weaknesses:**

**Strengths**

The problem setting is clear, and the proposal is a nice application of Finzi et al., ICML, 2021.

**Weakness**

Novelty and significance appear to be limited. Several studies have already investigated the symmetries in dynamical systems.
Especially, Finzi et al., ICML, 2021, on which the proposed method is based, already examined dynamical systems symmetric to continuous groups such as $O(2)$. Discrete symmetries have also examined in the references in Finzi et al., ICML, 2021. It is not clear what main difficulties were and how the proposed method resolved.

The morphological discrete symmetries in robots have also examined in Finzi et al., "Residual Pathway Priors for Soft Equivariance Constraints", NeurIPS, 2021, which is not cited in the present manuscript.

Also, the differences from previous studies (especially, Weissenbacher et al., 2022) are still unclear. Please introduce Limitations in the Appendix in more detail.


In Section3, the definition and expression about the velocity contain several abuses of notions. $TQ$ is often used to represent a tangent bundle (the space of the pair of position and velocity). $T_qQ$ is the notion for a tangent space (the space of the velocity) at point $q\in Q$. The map $\dot g$ accepts $q$ as an argument, but if so, its domain should be $Q$, not $TQ$. Nothing can map a tangent vector $\dot q$ independently from the position $q$. The map should be the pushforward by $g$ at the position $q$ and can be denoted by (but not limited to) $g^*_q$.



**Summary Of The Paper:**

This study focuses on the morphological symmetries in dynamical systems; for example, a bipedal robot is symmetric to the reflection. When applying data-driven method to such systems, the function is desired to have corresponding symmetries. To this end, this study uses the method proposed in ICML,2021 to build symmetric neural networks.


**Summary Of The Review:**

While the problem statement and the proposed method are clear, they seem combinations of existing studies. It is better to emphasize the novelty.

---

> ### Author Response · Authors · 2022-11-15
> **Response to reviewer tbqUdyamics**
>
> Dear reviewer tbqU.
>
> In this initial comment, we intend to address some of your concerns, doubts, and criticism of our work.
>
> * *"The problem setting is clear, and the proposal is a nice application of Finzi et al. (2021) EMLP [...] Novelty and significance appear to be limited. Several studies have already investigated the symmetries in dynamical systems. Especially, Finzi et al., ICML, 2021, on which the proposed method is based, already examined dynamical systems symmetric to continuous groups [...]. Discrete symmetries have also been examined in the references in Finzi et al., ICML, 2021."*
>
> As the reviewer points out our work can be considered an application of $\mathcal{G}$-equivariant NN (namely of Finzi. et al.(2021) EMLP. However, our contributions extend beyond being a specific implementation of $\mathcal{G}$-equivariant NNs. In fact, this project was originally intended to focus on learning locomotion control policies (for bipedal and quadrupedal robots) with $\mathcal{G}$-equivariant NNs powered by Finzi. et al (2021) EMLP EMLP framework. During the development of the project, we came to two conclusions:
>
> 1. Present literature, both in machine learning and dynamical systems, was lacking an analysis of the discrete symmetries of dynamical systems evolving in a euclidean space (e.g., flying/swimming/walking robots/animals/animated characters, among others) from the robotics/dynamical systems perspective. As stated in the introduction, the lack of framework for the **identification** of the symmetry group $\mathcal{G}$ and its representations in different vector spaces is the main reason why the research community does often not exploit these symmetries.
>
>     We strongly believe that our analysis of these discrete symmetries is our main contribution, as we provide novel insides that we hope will boost the exploitation of finite symmetry groups in robotics, computer graphics, experimental biology, and ML. Here, we briefly highlight the novelties of our analysis from the one done by Finzi. et al (2021) EMLP and other previous works in robotics/dynamical systems.
>     * **Identification of the symmetry group and its representations**: The biggest difficulty in applying $\mathcal{G}$-equivariant function approximators in dynamics is the identification of the symmetry group and the action representations in arbitrary vector spaces. We show that for floating-base dynamical systems with Configuration space $\mathcal{Q}\doteq\mathbb{E}_3 \times \mathcal{Q}_J$ evolving in $\mathbb{E}_3$ (or $\mathbb{E}_d$), discrete morphological symmetries $\mathcal{G} \in \mathcal{G}$ are easy to identify, by understanding that
>             (i) the action $\mathcal{G}$ should *imitate* an action of $\mathbb{E}_3$ (a rotation/reflection + translation in $3$ (or $d$) dimensions).
>             (ii) that Lagrangian $\mathcal{G}$-invariance (symmetries) imply $\mathcal{G}$-equivariant dynamics. Therefore, equivariant generalized mass Matrix. With these two constraints, we provide analytical means to find the action representations, for the case of rigid body dynamics. *(We have made substantial changes to section 4 to highlight this fact more explicitly)*
>    * **Symmetries of proprioceptive and exteroceptive measurements**: By assuming  $\mathcal{Q}\doteq\mathbb{E}_3 \times \mathcal{Q}_J$ , we show that any proprioceptive and exteroceptive measurements relevant for the evolution of the dynamics of the system can be augmented to their symmetrical equivalents solely with combinations of the representations in these two spaces. *(We have explicitly made this fact clear in section 4.1)*
>    * **Our analysis pays closer attention to dynamical systems theory (see new section 3)**: Although Finzi. et al (2021a/b) provide valuable theoretical and practical contributions to both dynamical systems and ML, these works have a stronger focus on ML (this is no criticism, but rather an acknowledgment). Note that:
>        * We provide a formal definition of symmetries as Lagrangian invariances and as dynamics equivariance. Clearly accounting for the symmetry-breaking nature of moving forces.
>        * Analyze the topology of the Configuration space $\mathcal{Q}$. And exploit it for symmetry identification.
>        * Study the case of rigid body dynamics, and provide insights on the morphological requirements of DMS.

---

> > ### Author Response · Authors · 2022-11-15
> > **Response to reviewer tbqU (2)**
> >
> > Continuing our conclusions of the [previous comment](https://openreview.net/forum?id=TBOFHtBariC&noteId=YEbRUxS8NBs)
> >
> > 2. Although novel, flexible, and highly valuable, the EMLP framework from Finzi. et al (2021) EMLP presented several limitations prohibiting its direct use in applications handling complex dynamical systems and real-world applications. Such as the static friction regime contact detection of Lin et al. (2021). *(We have covered these limitations in our changes to Section 5 and on **Supplementary D**)*
> >
> >    The core limitation is the inability to handle large dimensional spaces, due to the computational and memory complexities. For instance, for an equivariant layer with input dimension $n$ and output dimension $m$,  the computational complexity of finding the equivariant linear map basis $\mathbf{B}$ (which is quadratic $\mathcal{O}((mn)^2r^2)$ through the Krylov subspace method) and the memory complexity of $\mathbf{B} \in \mathcal{R}^{mn \times r} | r \leq mn$, become easily intractable for moderate $n$ and $m$. This limitation is openly discussed in the EMLP repository **README.md**, but regretfully not in the original paper.
> >     *(We added supp.table.1 which shows the high memory complexity of the matrix $\mathbf{B}$ for the Contact-CNN architecture, motivating for the sparse matrix representation)*
> >
> >
> >     Lastly, as we discuss in the introduction our contributions to machine learning are mainly practical, two of them being the handling of sparse representations and the algorithm for computing $\mathbf{B}$ in linear time, which allow us to construct $\mathcal{G}$-equivariant architectures of arbitrary size (for $\mathcal{G}$ a finite symmetry group). The other two contributions are (i) the optimal initialization of equivariant layers for any group $\mathcal{G}$ (which played a crucial role in reducing the variance across training seeds of equivariant architectures). And (ii), the analytical computation of the rank $r$ of $\mathbf{B}$. In the new manuscript, we explain in greater detail the implications of these contributions and the assumptions made. We invite you to have a look again at section 5.
> >
> >     PS: It is important to state that is our intention to merge our practical contributions with the EMLP framework of Finzi. et al (2021) EMLP. Making it a fundamental module in out DMS repository, which we hope will enable robotics/computer-graphics/experimental-biologist researchers to prototype easily $\mathcal{G}$-equivariant models for their experiments.

---

> > > ### Author Response · Authors · 2022-11-15
> > > **Response to reviewer tbqU (3)**
> > >
> > >  * *"Please introduce Limitations in the Appendix in more detail"*
> > >
> > > As suggested, we have moved the section on limitations to the appendix and extended it to provide extensive details on the two main assumptions (exact symmetries and generalized permutation matrix regular representations) that partially limit the applicability of our work.
> > >
> > > We apologize to the reviewers for the previous attempt to explain the limitations of our work, which, because of the page limit, lack the relevant information and attention to detail to clearly portray the reasons for our assumptions and their impact on the exploitation of DMS in robotics, computer graphics, biology and other fields of knowledge processing the dynamics of agents/bodies evolving in the euclidean space.
> > >
> > > * *"In Section3, the definition and expression about the velocity contain several abuses of notions [...]"*
> > >
> > > The reviewer is absolutely correct. We duly apologize for these errors & typos and thank the reviewer for its accurate correction and attention to detail. We have introduced several changes in the notation and definitions in Section 3, correcting this error and reducing notation complexity. We hope to hear your opinion on this version. :)

---

> > > > ### Comment · Reviewer_tbqU · 2022-11-17
> > > > **Thank you for response.**
> > > >
> > > > Thank you for your response, but please keep responses to the minimum necessary. Responses that are too long and too topical cause further confusion. If very detailed comparisons are required to show the novelty, then this indicates that the novelty is limited.
> > > >
> > > > > The biggest difficulty in applying -equivariant function approximators in dynamics is the identification of the symmetry group and the action representations in arbitrary vector spaces. We show that for dynamical systems with a trivial principal fiber bundle as configuration space evolving in (or ), discrete morphological symmetries are easy to identify, by understanding that (i) the action should \textit{imitate} an action of in the fiber (a rotation/reflection + translation in (or ) dimensions). (ii) that Lagrangian -invariance (symmetries) imply -equivariant dynamics. Therefore, equivariant generalized mass Matrix. With these two constraints, we provide analytical means to find the action representations, for the case of rigid body dynamics. (We have made substantial changes to section 4 to highlight this fact more explicitly)
> > > >
> > > > Yes, then the authors have found a nice target for applying EMLP; however, this seems to be only a special case of the work of Finzi et al.
> > > > I acknowledge the contribution in showing that the method of Finzi et al. can be very useful in areas such as robotics and the procedures for applying them.
> > > > However, I did not find any non-trivial contributions to ML, such as those that combined ML with dynamics or ML and symmetry, as the authors stated that *our contributions to machine learning are mainly practical*,
> > > > As such, I cannot strongly recommend that this paper be accepted.
> > > > I would gladly give this paper high marks if it were at a conference dedicated to robotics.

---

> > > > > ### Author Response · Authors · 2022-11-17
> > > > > **on the value of contribution for ICLR (DL applications) (DL practical contributions)**
> > > > >
> > > > > Dear reviewer (tbqU).
> > > > >
> > > > > * *" However, I did not find any non-trivial contributions to ML, such as those that combined ML with dynamics or ML and symmetry, as the authors stated that our contributions to machine learning are mainly practical, As such, I cannot strongly recommend that this paper be accepted. I would gladly give this paper high marks if it were at a conference dedicated to robotics."*
> > > > >
> > > > > We would like to point out the [description of the objectives and topics covered by ICLR](https://iclr.cc/):
> > > > > >*"ICLR is globally renowned for presenting and publishing cutting-edge research on **all aspects** of deep learning used in the fields of artificial intelligence, statistics, and **data science**, as well as important *application areas* such as **machine vision**, **computational biology**, speech recognition, text understanding, **gaming**, and **robotics**."*
> > > > >
> > > > > We believe the description should persuade you of the value of our contribution to ICLR since:
> > > > > -  *"all aspects of DL"* -> including practical aspects of DL
> > > > > - Among the [relevant topics explored at the conference](https://iclr.cc/) there is:
> > > > >     - applications in audio, speech, **robotics**, neuroscience,  **biology**, or **any other field**
> > > > >     - **implementation issues**, parallelization, **software platforms**, **hardware**
> > > > >     - representation learning for planning and reinforcement learning
> > > > >     - learning representations of outputs or states
> > > > >
> > > > > -------------------------------------------------
> > > > >
> > > > > * *"If very detailed comparisons are required to show the novelty, then this indicates that the novelty is limited."*
> > > > >
> > > > > We took the effort to produce very detailed responses to portray without a doubt why your concerns might be misguided. We have updated answers to be more concise, but here we summarize briefly again for you:
> > > > >
> > > > > 1. [Comparison with "Koopman q learning" Weissenbacher et al 2022 ](https://openreview.net/forum?id=TBOFHtBariC&noteId=h4T0Lq9Q4p): There is little to no overlay of ideas between Weissenbacher et al 2022 et al. and our proposal. The papers have completely different assumptions and objectives. (Read the response for details). Note that by knowing the topology and the symmetries of the dynamics in minimal coordinates $\mathbf{q}$ (i.e. DMS contribution), you would (without a doubt) want to learn Koopman observable mappings that preserve the structure (topology and symmetries), and would like to impose the $\mathcal{G}$ symmetries on the Koopman operator. I.e., DMSs will undoubtedly lead to the learning of better Koopam operators (one of the objectives of Weissenbacher), by analytically determining the symmetries, instead of approximating them.
> > > > >
> > > > > 2. [Comparison with Residual Pathway Priors Finzi et al. NeurIPS, 2021](https://openreview.net/forum?id=TBOFHtBariC&noteId=KNtvuR-rjmb): Please read the limitation section of (Finzi et al. NeurIPS, 2021). It's a good description of the value of our theoretical contributions. We invite you to read Supplementary D (of the new version of the manuscript) for understanding our practical contributions.
> > > > > We would also like you to not forget about of contribution to the **optimal initialization of equivariant layers**, and the analytical determination of $r$ (the rank of the space of equivariant linear maps).

---

> > > > > > ### Comment · Reviewer_tbqU · 2022-11-24
> > > > > > **Yes**
> > > > > >
> > > > > > I know the standards, and this is why I put 5, not 3. Simply speaking, the novelty and significance were marginally below the required level.

---

> > > > > > > ### Author Response · Authors · 2022-11-29
> > > > > > > **Further Comments**
> > > > > > >
> > > > > > > Dear reviewer tbqU,
> > > > > > >
> > > > > > > We would like to ask, if possible, for more details on what requirements and properties you consider to judge each of our contributions as valuable/not valuable, above/below the required level. This could allow us to better understand your concerns, in order to produce better work and better argue with your assumptions, in case we recognize a misconception.
> > > > > > >
> > > > > > > If it is not too much to ask we would like to know your novelty assessment argumentation for:
> > > > > > >
> > > > > > > 1. The practical implementation contributions described in Section 5 and especially in the new **Supplementary D** (added to ease your initial review concerns). Which allows the use of $\mathcal{G}$-equivariant (in the case of DMS) in real-world applications. This includes:
> > > > > > >     * Optimal initialization of equivariant layer trainable parameters.
> > > > > > >     * Analytical estimation of the number of trainable parameters $r$ after imposing equivariant constraints.
> > > > > > >     * Implementation of Sparse representations and Sparse computation of the basis of equivariant linear maps $\mathbf{B}$. (see the practical value of this contribution in Supplementary Table 1)
> > > > > > >     * Linear time algorithm proposition and implementation for computation of $\mathbf{B}$ in linear time for DMS groups.
> > > > > > >  2. The proposed concept of Discrete Morphological Symmetries, as a generalization of the Reflection symmetry to higher order finite groups (Section 4). Including:
> > > > > > >     * The realization of the equivariant nature of the Generalized Mass Matrix as an identifying property of symmetric dynamical systems (Eq. 4)
> > > > > > >     * The realization that any data structure symmetry action representation can be built purely with the action representations on $\mathbb{E}_d$ and $\mathcal{Q}_J$ of the symmetry group (easing what was previously an [error-prone](https://openreview.net/forum?id=TBOFHtBariC&noteId=KNtvuR-rjmb) process). Along with the tutorials in (Supp Fig 5, Supplementary E.3.1 and E.4.1)
> > > > > > >     * The analysis of DMS for rigid body dynamics (section 4.2). Useful for robotics, computer graphics, and computational biology. As justified in Supplementary A.
> > > > > > >   3. [The repository with the code](https://anonymous.4open.science/r/RobotEquivariantNN/README.md) to reproduce our experiments and to prototype NN-based architectures processing proprioceptive and exteroceptive measurements of equivariant rigid-body dynamical systems. Featuring:
> > > > > > >      * An expanding library of several dynamical systems with their DMS groups already identified. Currently featuring robotic systems but soon expanded to common agents used in computer graphics, and computational biology.
> > > > > > >      * A object-oriented implementation of the representations on DMS and the representations $\mathbb{E}_d$ and $\mathcal{Q}_J$
> > > > > > >
> > > > > > >
> > > > > > >
> > > > > > > Lastly, we like to thank you for the time you invested and (hopefully will invest) in this reviewing process. Having a clearer idea of your thought process will undoubtedly increase the constructive nature of your assessments.

---

> ### Author Response · Authors · 2022-11-15
> **Comparison with Residual Pathway Priors Finzi et al. NeurIPS, 2021**
>
> * *"The discrete morphological symmetries in robots have also been examined in Finzi et al., "Residual Pathway Priors for Soft Equivariance Constraints", NeurIPS, 2021, which is not cited in the present manuscript."*
>
> We were unaware of this new contribution from Finzi et al. We thank the reviewer for pointing out this valuable contribution to us. Since Residual Pathway Priors are a clear methodology for approaching the problem of approximate $\mathcal{G}$-equivariance (as Wang et al. (2022)), we have duly cited them in the limitations appendix section, where we discuss the assumption of exact symmetries. Furthermore, we have added it to the relevant literature in the introduction of the manuscript, among the other previous works that have exploited specific instances of DMSs in the past with beneficial results.
>
>   Regarding the differences with Finzi et al. (2021) RPP  specifically, we note that although most symmetries discussed in Finzi et al. (2021) RPP are specific cases of DMSs, there are several inconsistencies in the work that we believe arise from a not-so-detailed analysis of the symmetries from a dynamical system perspective (again no criticism here, as the paper main and highly valuable contribution is aimed at Machine Learning). Among these inconsistencies we have that:
>
> *  Ant dynamical system has exact and not approximate Klein four symmetry. This can be concluded following the definition and analysis of DMSs. Since the base of the robot has the required symmetric mass distribution (allowing it to imitate reflections of space w.r.t to two principal axes of inertia) and symmetric constraint manifolds $\mathcal{Q}_J$ and $\mathcal{T}_q\mathcal{Q}$.
> *  The Half-Cheetah dynamical system has no exact nor approximate symmetry from the lens of DMS following the definition in Section 3. Since this is a case of rigid-body dynamics evolving in $\mathbb{E}_2$, it is not difficult to identify that the only DMSs that this system can have is one which imitates a reflection of space w.r.t the vertical axis. Since the base body of the robot does not have a symmetric mass distribution (and the legs have a different number of degrees of freedom, implying different dynamic roles in locomotion), in the author's opinion (from the lens of legged locomotion control) there is no relevant kinematic nor dynamic symmetry to exploit.
>
>
> **Note that we specifically address the **limitations stated in Finzi et al. (2021)** RPP:**
>
> >  *"Using RPP-EMLP for the state and action spaces of the Mujoco environments required **identifying the meaning** of each of the components in terms of whether they are scalars, velocity vectors, joint angles, or orientation quaternions, and also which part of the robot they correspond to. **This can be an error-prone process**. While RPPs are fairly robust to such mistakes, **the need to identify components makes using RPP more challenging than standard MLP**. Additionally, due to the bilinear layers within EMLP, the Lipschitz constant of the network is unbounded which can lead to **training instabilities** when the inputs are not well normalized. We hypothesize these factors may contribute to the training instability we experienced using RPP-EMLP on Humanoid-v2}".* Finzi et al. (2021) RPP
>
>    Although our work was developed without the knowledge of the existence of Finzi et al. (2021) RPP, it's important to highlight that our work addresses these limitations by (i) Mitigating the errors in the identification of symmetry groups and representations for dynamical systems (Section 3) of rigid-body systems (Section 4.2) and data related to the dynamics (section 4.1). (ii) By stabilizing the training of equivariant layers with an optimal initialization of the trainable parameters. (Section 5, and Supplementary D)

---

> ### Author Response · Authors · 2022-11-15
> **Comparison with "Koopman q learning" Weissenbacher et al 2022**
>
>  *  *"Also, the differences from previous studies (especially, Weissenbacher et al., 2022) are still unclear."*
>
> The core difference in our work from Weissenbacher et al. 2022 relies on the assumption of the nature of the symmetries of the dynamical systems studied and on **how to determine the symmetry action representations**.
>
>  * **On the nature of symmetries:** Weissenbacher et al. (2022), only study continuous symmetries of dynamical systems related to the actions of the Lie Group $\mathbb{E}_d$ in which the system evolves (see definition 3.1 in Weissenbacher et al. 2022). Whereas our work makes a clear distinction between the continuous symmetries and the **discrete symmetries** (related to Discrete Morphological Symmetries. Section 3) that are a consequence of specific morphological properties of the dynamical model. Our work **focuses on the latter, i.e. in DMS** (finite group symmetries. Section 4.0), as these are **frequently ignored and left unexploited** in the fields of robotics, computer graphics/vision, and biology/biomechanics.
>
> *    **How to determine the action representations:** Weissenbacher et al. (2022) propose to find the symmetry action representations as equivariant actions in the space of observables of a finite-dimensional **approximation** of the Koopman operator, which is learned **with expert-controlled trajectories** of the dynamics of the system:
> >*" However, symmetry maps are hard to come by. [...] we provide a principled approach on how to derive this symmetry maps for control system using Koopman theory"* (Weissenbacher et al. 2022).
>
> Our approach **does not** rely on the approximation of a Koopman operator, **nor requires expert trajectories** for the determination of the symmetries. Instead, we state that symmetry action representation (and thus the Group $\mathcal{G}$) can be discovered as actions to which the generalized mass matrix $\mathbf{M}(\mathbf{q})$ is $\mathcal{G}$-equivariant (eq.(4) and section 4.2). In the case of rigid-body dynamics, we provide analytical methods for finding these representations.  We do **assume knowledge of the nature/structure of the generalized mass matrix** of the system, which happens to be true for robotics, rigid-body dynamics, and any modeled dynamical system.

---

### Official Review · Reviewer_Yn9H · 2022-10-23

**Confidence:** 5
**Correctness:** 4
**Technical Novelty And Significance:** 3
**Empirical Novelty And Significance:** 3
**Recommendation:** 5

**Clarity, Quality, Novelty And Reproducibility:**

suggestion: to simplify the second sentence in Section 4, which is a 7 lines sentence. It is the main point of the paper, which could be more readable. I will reconsider my score, provided the authors address my concerns in "Weaknesses".



**Strength And Weaknesses:**

Strength:
1. DMS is an interesting idea
2. fabulous representation of the background

Weaknesses:
As a whole this paper is good. It does contribute to the field and combines different advanced approaches. My main concerns are
1. that the DBS itself is a tiny part of the paper, which is built on and plugged in to the existing methods.
2. DBS can not be applicable to many dynamics, which is mentioned also in "Imitating reflections of space requires symmetric rigid bodies and/or modularity"
3. DBS requires significant domain knowledge (when and how to mimic the original symmetry). there is no general principle how to calculate it in an arbitrary dynamics. it is not unique, as there can be multiple reconfigurations mimicking the original symmetry.
4. the experiments are limited to supervised learning, even though the author method in the text uncontrollable and consolable trajectories, which alluded there would be application to control problems.





small comments:
typos: "Kinematic three"," an angle $\theta$ unfeasible".

**Summary Of The Paper:**

this paper proposes to mimic physically unrealizable symmetries such as reflections, by reconfiguration of dynamics such as left/right legs shift in bipedal.

**Summary Of The Review:**

as a whole it is a good paper. the idea is interesting. the novelty is limited. the applicability is limited.

---

> ### Author Response · Authors · 2022-11-15
> **Reponse to reviewer Yn9H**
>
> Dear reviewer Yn9H
>
> In this comment, we will try to address all of your concerns. We like to thank you for pointing out the lack of clarity on the flexibility and practicality of the concept of DMSs for real-world applications. This allows us to recognize that the previous manuscript failed to convey properly the main idea of out work. Namely, that DMS is a flexible and highly applicable framework for data-driven applications in Robotics, Control, Computer Graphics, and Experimental Biology.
>
> First, we want to comment on the strengths mentioned by the reviewer:
>  * *"DMS is an interesting idea"* and *"Fabulous representation of the background"*
>
> We sincerely appreciate that you find our proposal interesting. We are very excited about its applicability. Regarding the figure on the background section (the K4 symmetry on the real robot and on the example equivariant NN), we would like to invite you to see the [3D animation](https://imgur.com/a/30M0sym) of the K4 symmetry of Solo and of the [humanoid Atlas](https://imgur.com/a/A6RBRk7) and [bipedal Bolt](https://imgur.com/a/CYj8wTQ). These last two with $\mathcal{G}=\mathcal{C}_2$. All these animations were made with exactly the same script robot_visualization.py available in the anonymized repository.
>
> To showcase the flexibility of the DMS formalism we are planning to release our repository with a large library of robotic platforms with their respective symmetry groups (probably the majority of robots listed on [robot_descriptions_package](https://github.com/robot-descriptions/awesome-robot-descriptions#gallery)). Including single and double arm manipulators, bipedal and quadrupedal robots, and drones. So I invite you, before the final deadline, to check our anonymized repository for more examples and cool animations.
> _________________________________
> Now we address the weaknesses highlighted by the reviewer.
>
>
>   *  *"As a whole this paper is good. It does contribute to the field and combines different advanced approaches. My main concerns are that the DBS itself is a tiny part of the paper, which is built on and plugged into the existing methods"*.
>
> We would like to invite you to review again sections 3 and 4. Where we have introduced several changes to highlight the main point of the paper. Additionally, section 4. (Discrete Morphological Symmetries) has been extended extensively to describe the main contributions of the DMS formalism. That is:
>
>    * That the concept of DMS is a **generalization** of the common reflection symmetry to higher order finite symmetry groups $|\mathcal{G}| \geq 2$. (section 4.0)
>    * That **identifying** the Symmetry Group $\mathcal{G}$ and the action representations on euclidean space/group $\mathbb{E}_d$ and joint-space $\mathcal{Q}_J$, opens the door for the augmentation of any proprioceptive and exteroceptive measurement related to the evolution of the dynamics of the system. (section 4.1)
>    * Lastly, we analyze the case of rigid-body dynamics (section 4.2), for which we conclude that:
>         * A DMS imposes several constraints on the morphological properties of the bodies of the system. Which in practice are **satisfied** by a **large variety** of rigid-body floating base systems.
>         * A justification on why the **morphological restrictions** are useful for the **identification** of the symmetry group $\mathcal{G}$.
>
> We also provide Step-by-step indications on how to identify DMS for rigid-body systems.

---

> > ### Author Response · Authors · 2022-11-15
> > **Reponse to reviewer Yn9H (2) - Applicability and practicality of DMSs**
> >
> > [Continuing ...]
> >
> >   * *"[DMS] can not be applicable to many dynamics, which is mentioned also in "Imitating reflections of space requires symmetric rigid bodies and/or modularity"*
> >
> > As it is better justified in *Supplementary A: Applications of DMS* (please review) the **majority** of animal species have bilateral symmetry (this also translates to the majority of animated characters), implying that there is a large number of dynamical systems that, to the least, fulfill the requirements to having a reflection symmetry (even in this case the exploitation of the symmetry is recommended).
> >
> > In the case of robotics, as symmetries are more precise, the reflection symmetry is also typical, and there exist several robot designs already featuring the requirements for higher-order DMS $|\mathcal{G}| \geq 2$ such as the robot Solo and other quadrupeds, most quadcopters/hexacopters, most Hexapos, among others.
> >
> > I believe the notion that DMS is not applicable to a large number of dynamical systems is caused by confusion over the claims of the original manuscript and an excessive focus on the reflection case. We took a lot of effort into explaining the constraints on kinematic and dynamic parameters in the new section 4.2. Where we hope it's clear that **in practice, the symmetric mass distribution constraint, is only applied to *unique* bodies in the kinematic tree (like the base of the system)**. While **non-unique bodies (i.e., bodies replicated in multiple subchains of the kinematic tree) are not constrained by this symmetric mass distribution constraint**, since the replication itself can do the job.
> >
> > To stretch this point, notice that as we explain in sections 4.2 and 4.3, **the symmetries in the mass distribution of the base of the robot (or other unique bodies) are good indicators of what Euclidean transformation could the system imitate. And thus, of what might be the DMS group $\mathcal{G}$**
> > ______________
> >
> >  * *"[DMS] requires significant domain knowledge (when and how to mimic the original symmetry). there is no general principle on how to calculate it in arbitrary dynamics. it is not unique, as there can be multiple reconfigurations mimicking the original symmetry."*
> >
> > We tried to specifically address this concern with sections 3.0, 4.2, and 4.3. (especially section 4.3 **where we describe the *"general principle"* for the case of rigid-body dynamics**. Please review them). Additionally, we Introduced **tutorials and examples** on the identification of the symmetry group and group action representation in supp.fig.5 and Supplementary E3.1 and E4.1.
> >
> > In summary, the fact that any dynamical system with a symmetry group $\mathcal{G}$ (finite or continuous) is required to have equivariant dynamics implies that it also must have an equivariant generalized Mass matrix *function* $\mathbf{M}(\mathbf{q})$ (this fact was a bit hidden but implicitly expressed in the dynamics $\mathcal{G}$-equivariance equations of the original manuscript). In the new manuscript, we introduced eq. (4) to explicitly convey this fact to the reader.
> >
> > Since $\mathbf{M}(\mathbf{q})$ is known for most dynamical systems (for which we have an approximate model), the equivariance of this function becomes a useful pathway for the identification of the system symmetries.
> >
> > We show in the paper that once we assume rigid-body dynamics, the $\mathbf{M}(\mathbf{q})$ has a particular structure that allows us to translate the equivariance constraints of $\mathbf{M}(\mathbf{q})$ to constraints in the positional and rotational Jacobians of the system bodies (and therefore to the kinematic and dynamic parameters of the system (section 4.2)). A similar analysis can be done for soft robot bodies, or other system architectures, where we have approximate knowledge of $\mathbf{M}(\mathbf{q})$.
> >
> > We limit ourselves to rigid-body dynamics as is the most common type of dynamics in Robotics, Computer Graphics, and Experimental Biology.

---

> > > ### Author Response · Authors · 2022-11-15
> > > **Comment on the non-uniqueness of symmetry transformations imitating an Euclidean transformation.**
> > >
> > > Following on your claim *"as there can be multiple reconfigurations mimicking the original symmetry"*.
> > >
> > > The interesting property of identifying the finite symmetry group $\mathcal{G}$ is that any group in group theory is known to be generated by a set of *[group generators](https://mathworld.wolfram.com/GroupGenerators.html#:~:text=is%20a%20set%20of%20group,powers%20of%20a%20single%20generator.)* actions. That means that any action in the group can be made as a combination of some of the generators of the group.
> > >
> > > As an example consider a Hexagon (or a Hexapod, to make it cooler) you have six different symmetric configurations related to $60^\circ *n| n \in \mathbb{N}$ rotations. Therefore 6 actions in the group, but in reality only one generator (which when multiplied multiple times generates the rest of the actions). This knowledge can be exploited specifically in point (2) of Section 4.3. That is, following our example if we identify the base inertia is invariant to $60^\circ *n| n \in \mathbb{N}$ rotations, we could understand that the possible largest group that the system can imitate is the cyclic group $\mathcal{C}_6$. If this is the case, finding the representation of the action $g$ that imitates the Euclidean transformation $\overline{g}\equiv 60^\circ$, will signify, finding the representation of the generator of $\mathcal{C}_6$ on $\mathcal{Q}$.
> > >
> > > I have to admit that this is a very interesting topic to think over for some time. In full honesty, I have not find empirically yet a situation where this occurs beyond the trivial classes of cyclic or dihedral groups where a little knowledge of group theory (or brute force since the order of the groups is never large **for rigid-body** systems) can resolve the issue. If you have any particular example in mind, let me know and we can discuss it. Note that in none of the examples of the paper this situation occurs.
> > >
> > > Although I would like to think over this more deeply and probably produce a new section, I am afraid I have run out of space and time to do it. However, I do not think this is an aspect of the paper that should affect negatively your opinion of our work. Rather, I would say that if you find this comment interesting, then perhaps it is a good indicator that there is still a lot more to explore on this idea.
> > >
> > > PS: An interesting fact that you might already know. When imposing constraints on a $\mathcal{G}$-equivariant NN, the constraints imposed are given only by the generators of the group (Finzi et al. 2021 EMLP).

---

> ### Author Response · Authors · 2022-11-16
> **Experiments limited to supervised learning not control (reinforcement learning)**
>
> *  *"the experiments are limited to supervised learning, even though the author's method in the text uncontrollable and consolable trajectories, which alluded there would be application to control problems"*
>
> As we mentioned in [this response to the reviewer (tbqU)](https://openreview.net/forum?id=TBOFHtBariC&noteId=YEbRUxS8NBs), our original project idea was directed towards learning control locomotion policies with $\mathcal{G}$-equivariant NN for a large variety of locomoting systems. Much in the spirit of Ordonez-Apraez et al. 2021 and Abdolhosseini et al. 2019, but experimenting with a **large number of animated characters and robotic systems**, with **varying morphologies** (different numbers of legs and architectures).
>
> During the development of this project, we faced the problem of **identifying the symmetry action representations for the proprioceptive and exteroceptive measurements of the data** (which are usually the input to model-based/model-free control policies). Then we **recognized that several platforms have more than the reflection symmetry** explored in Ordonez-Apraez et al. 2021 and Abdolhosseini et al. 2019 (and other works cited in the introduction), which further complicated the problem of identification of not only one action representation but, for some systems, **more than one action representation**. *This is a problem recognized and faced by other works (see [last point of this response](https://openreview.net/forum?id=TBOFHtBariC&noteId=KNtvuR-rjmb)).*
>
> Working on solving this problem we developed the concept of DMSs which as mentioned in section 4.0 (of the new manuscript) can be understood as a **generalization of the phenomena of the *reflection* symmetry $\mathcal{G}=\mathcal{C}_2$ (present in most animals/robots/animated characters), to scenarios where the finite symmetry group has a larger order  $|\mathcal{G}| \geq 2$ (meaning that a DMS action can imitate reflections/rotations/translations)**. Furthermore, as explained in section 4.1 **DMS solves the problem of identifying the representation of proprioceptive and exteroceptive measurements** by recognizing that the representations of these measurements can be constructed solely with combinations of the action representations in $\mathbb{E}_d$ and $\mathcal{Q}_J$ (i.e. the symmetry representation on the space in which the system evolves, and the symmetry representation in joint-space).
>
> We decided to **make DMS the focus of our work since we firmly believe it is a relevant contribution to various fields (and a fundamental preamble to address projects such as the originally intended)**. To make DMS *reproducible* and more importantly exploited by other researchers, we tackle **several of the practical limitations** that the state-of-the-art methods for constructing  $\mathcal{G}$-equivariant NN have. Which increases the difficulty of the exploitation of DMS in robotics and computer graphics. Note that our reference in the introduction *" Notably, Zinkevich & Balch (2001) proved that Markov Decision Processes with state symmetries have symmetric
> optimal value and policy functions."* is intended to make clear that learning methods on MDPs (and thus control) are **a direct application of symmetry exploitation**.
>
> **Regarding RL experiments**: since most learning-based control methods can be cast as supervised learning problems, we decided that the best way to portray the functionality of our work is by addressing supervised learning experiments. We deliberately chose a **real-world relevant robotics application** as an experiment to showcase the potential that exploiting these symmetries has in practice. Additionally, notice that several works have explored the impact of symmetry exploitation in reinforcement learning applications. Namely:
> - On the impact on sample efficiency, generalization, and final performance of model-free RL algorithms you have (Finzi et al. 2021 RPP, Ordonez-Apraez et al. 2021, Abdolhosseini et al. 2019, Van der Pol et al., 2020)
> - On the impact of symmetry exploitation for learning robust/realistic locomotion policies with model-free RL you have (Ordonez-Apraez et al. 2021, Abdolhosseini et al. 2019)

---

### Official Review · Reviewer_vevv · 2022-10-25

**Confidence:** 3
**Clarity, Quality, Novelty And Reproducibility:** See above.
**Correctness:** 4
**Technical Novelty And Significance:** 4
**Empirical Novelty And Significance:** Not applicable
**Recommendation:** 6

**Strength And Weaknesses:**

Pros:
+ The paper is well-written and easy to read.
+ The literature review is comprehensive. The presented results are well-positioned in the existing works.
+ The presented theoretical foundations for robotics systems are novel and make sense. The authors also present detailed proofs.

Cons:
- The presented results lack a clear motivation. The authors are encouraged to discuss what the presented results can contribute to. The paper says “addressing the memory
and computational complexity drawbacks of using this type of architecture.” However, no theory or experiments are give to support this. Overall, I am concerned about the impact of this paper.


**Summary Of The Paper:**

This paper studies the discrete symmetries of robotics systems. The authors present a set of results: (1) the identification of the system’s morphological symmetry Group G, (2) the characterization of how the group acts upon the system state variables and any relevant measurement living in the Euclidean space, and (3) the exploitation of data symmetries through the use of G-equivariant/G-invariant Neural Networks. However, the motivations and insights are not clear to me.

**Summary Of The Review:**

I find the results are novel and interesting but I am concerned about the motivation and impact of the results.

---

> ### Author Response · Authors · 2022-11-15
> **Response to reviewer vevv**
>
> Dear reviewer vevv,
>
> In this response we intend to address all of your concerns:
>
> * *"The presented results lack a clear motivation. The authors are encouraged to discuss what the presented results can contribute to."*
>
> Reviewer vevv was not the only one to have doubts over the motivation for DMSs or their practicality. In retrospect, we now see that our previous manuscript lacked the clarity to expose both the motivations and the impact of DMSs. To address this we have introduced several changes in the paper:
>
> We have introduced several modifications to sections 3 and 4 to highlight that the objective of our work is to provide a pathway for the identification of discrete symmetries of dynamical systems and their exploitation in data-driven applications. As it is stated in the introduction
>
> >*"[...], exploiting the inductive bias of morphological symmetries is not a widespread technique in the research community. We attribute the scarce adoption of these techniques to the lack of a unifying theoretical and practical framework, allowing to identify different morphological symmetries in arbitrary dynamical systems
> and efficiently and conveniently exploit them in data-driven applications. [...] This work takes a step towards this unifying framework [...]"*
>
> Namely, we have:
>
>    *  Updated the list of contributions in the introduction to better reflect the motivation of the paper.
>    *  We modify our formulation, vocabulary, and notation to highlight the fact that our contributions are not limited to robotics, but rather to any knowledge area where floating-base dynamical systems and specially rigid-body dynamics are used. I.e., in Computer Graphics, Experimental Biology, Control, and Robotics.
>    *   Introduced sections 4.1, 4.2, and 4.3 to highlight the versatility of the concept of DMSs in the identification of finite symmetry groups and their representations for the state variables and for proprioceptive and exteroceptive measurements. Facilitating the process of data augmentation and $\mathcal{G}$ equivariant function approximation.
>    * Introduced **tutorials and examples** on the identification of the symmetry group and group action representation in supp.fig.5 and Supplementary E3.1 and E4.1.
>    *  We added Supplementary A: **APPLICATIONS OF DISCRETE MORPHOLOGICAL SYMMETRIES**. Where we provide a detailed perspective on the range of possible applications of DMS in the fields of Computer Graphics, Robotics, Control, and Experimental Biology.
>
> _______________________
>
> *  *"The paper says “addressing the memory and computational complexity drawbacks of using this type of architecture.” However, no theory or experiments are given to support this"*
>
> To clarify the limitations that prohibited the construction of $\mathcal{G}$ equivariant NN for real-world applications we have introduced the Supplementary D: **EFFICIENT CONSTRUCTION OF G-EQUIVARIANT NNS FOR DISCRETE MORPHOLOGICAL SYMMETRY (DMS) GROUPS G.** Where we discuss the memory and computational complexity of the framework of Finzi. et al 2021 EMLP
>
> **On the memory complexity:** In supplementary D, we have introduced Supplementary table (1), which displays the prohibiting memory complexity of the matrix $\mathbf{B}$ for the architecture Contact-CNN (Lin et al. 2021), for which the widest layer required 1.50PBytes of memory to store the matrix. This means that without sparse matrix representations for this architecture is impossible to impose equivariant constraints.
>
> **On the computational complexity:** In supplementary D, we discuss the intractability to apply the methods for finding the matrix $\mathbf{B}$ , with quadratic computational complexity, for layers with reasonably large input-output space dimensions. Similarly as in the case of memory complexity, applying these methods to the biggest layer of Contact-CNN (Lin et al. 2021) signifies weeks of computing time with standard hardware. We believe a table with figures comparing the quadratic and linear computational complexities is unnecessary, but it can be added to the supplementary D if the reviewer **vevv** deems it necessary.  Note that the linear time algorithm for computing $\mathbf{B}$ is on our repository, as it is fundamental in the reproducibility of the results.
>
> PS: The reviewer tbqU had similar concerns. Although we answer here briefly. If you have time, we invite you to read the response to reviewer tbqU also (https://openreview.net/forum?id=TBOFHtBariC&noteId=vda0-dKSXF).

---

### Decision · Program_Chairs · 2023-01-20

**Decision:**

Reject

**Justification For Why Not Higher Score:**

The paper lacks focus

**Justification For Why Not Lower Score:**

n/a

**Metareview: Summary, Strengths And Weaknesses:**

The final reviewer scores are 5 (tbqU), 5 (Yn9H), 6 (vevv), putting this paper in borderline territory. Unfortunately reviewers Yn9H and vevv have not engaged in discussions. Therefore I have carefully read the paper and all responses, leading me to conclude the following:

Reviewers agree the paper is well written and gives an excellent introduction to the relevant background, much of which will be less familiar to the ML community than to physicists and roboticists. Personally, I found the paper interesting, though somewhat challenging due to (to me) somewhat unfamiliar background material. The analysis of discrete morphological symmetries in robotics systems / dynamical systems is one of the main contributions of the paper. The experimental results show decent performance, though they don't seem to reflect a major improvement.

Reviewer vevv mentions lack of motivation as the main weakness. The updated manuscript has made some improvements in this regard.

Reviewer Yn9H mentions:
- "DBS can not be applicable to many dynamics": as noted in the paper, many existing robots do satisfy some DBS.
- "DBS requires significant domain knowledge", "there is no general principle how to calculate it in an arbitrary dynamics". The updated paper does try to address this, though it remains true that finding DMSs requires domain knowledge and calculating by hand. This is probably more acceptable to roboticists than ML researchers.
- "the experiments are limited to supervised learning" [and don't include control]. I agree it would be nice to see such experiments.

Reviewer tbqU is mainly concerned about novelty relative to Finzi et al., ICML 2021, where a method for computing linear layers of equivariant networks for arbitrary groups is proposed. Indeed this paper does not introduce a new method for equivariant networks, though some novel tricks are introduced, e.g. for computing the basis B for sparse representations (this could be useful in other applications), though unfortunately these contributions are scattered throughout the paper and not studied in detail.

tbqU notes "I would gladly give this paper high marks if it were at a conference dedicated to robotics.". I conclude that this paper makes modest ML/equivariance-related contributions, and additionally makes a robotics-specific contribution, both of which are relevant to ICLR.

Since this papers is cross-disciplinary and does a lot of things at once, it is a bit hard to pinpoint exactly what its key contribution is, and requires an above-average cognitive effort to read. Some methodological contributions are made that could be turned into a paper on their own, if they were unpacked / explained in more detail, analyzed, and evaluated by rigorous experiments. Instead they are mentioned as an aside, without in-depth analysis and evaluation, which diminishes their value to the research community. The main analysis of DMSs will be straightforward for roboticists or physicists familiar with Langrangians and their symmetries, but may be hard to follow for ML researchers. Finally, the contribution is not so much a learning algorithm or architecture, but rather an analysis that requires explicit calculations by hand for each robot it is to be applied to, which will not seem very appealing to most ML researchers. Some of these issues are inherent to cross-disciplinary work, and in my view should not be reasons for rejection. Other issues could be better addressed by writing a longer journal or tutorial version (or splitting the paper), and I encourage the authors to do so.

In summary: several reviewer comments have been addressed. In my view the paper topic is relevant to ICLR. However, the paper is lacking in focus, and its main contribution is a bit hard to pinpoint, and the paper may be hard to read and in some ways unsatisfying to both roboticists and machine learning researchers. Still it's an insightful analysis with some promising early results, so I encourage the authors to continue to improve the research and publish it, perhaps in a longer format, at another venue.

**Summary Of Ac-Reviewer Meeting:**

I tried to set up a meeting but only 1 reviewer responded